# The effect of giant lateral collapses on magma pathways and the location of volcanism

Francesco Maccaferri[1], Nicole Richter[1] & Thomas R. Walter[1]

Flank instability and lateral collapse are recurrent processes during the structural evolution of volcanic edifices, and they affect and are affected by magmatic activity. It is known that dyke intrusions have the potential to destabilise the flanks of a volcano, and that lateral collapses may change the style of volcanism and the arrangement of shallow dykes. However, the effect of a large lateral collapse on the location of a new eruptive centre remains unclear. Here, we use a numerical approach to simulate the pathways of magmatic intrusions underneath the volcanic edifice, after the stress redistribution resulting from a large lateral collapse. Our simulations are quantitatively validated against the observations at Fogo volcano, Cabo Verde. The results reveal that a lateral collapse can trigger a significant deflection of deep magma pathways in the crust, favouring the formation of a new eruptive centre within the collapse embayment. Our results have implications for the long-term evolution of intraplate volcanic ocean islands.

[1] German Research Centre for Geosciences (GFZ), Potsdam, 14473, Germany. Correspondence and requests for materials should be addressed to F.M. (email: francesco.maccaferri@gfz-potsdam.de)

Especially tall and active volcanoes are prone to flank instability which may lead to failure and sector collapse[1–3]. Sector collapses are rather common during the evolution of a volcanic edifice, and may happen at any volcano[4]. Two prominent historical examples are Mount St. Helens, 1980, and Kobandai volcano, 1888, which removed 2.8 and 1.5 km$^3$ of rocks, respectively[5]. However, giant lateral collapses with volumes that may exceed 100 km$^3$ have been identified mainly at hot-spot ocean islands[3,6]. Lateral collapses might be preceded by stages of slow spreading or rifting processes[7].

Flank failures represent one of the most prominent sources of a large-scale volcanic hazard at ocean island volcanoes and have therefore been the subject of numerous studies[8,9]. For instance, submarine mapping surveys reveal deposits of about 70 giant landslides that occurred at the Hawaiian Islands, and about 20 events at the Canary Islands[3,10,11]. Furthermore, previous studies showed that flank collapses are recurrent at similar locations[9,12–15] and often decapitate a volcano's plumbing system[16]. Lateral collapses seem to occur more frequently when volcanoes are active[11], and their interaction with magmatic activity has been addressed by several studies (cf., McGuire[2], and the references therein).

Also, the idea of a mechanical effect of magmatic intrusions (dykes) on the stability of the volcanic edifice has been proposed and was subject to previous research[8,17,18]. It was shown that dyke intrusions may interact with decollement faults through the stress changes induced by the dyke inflation[9,17,19]. These interactions are sometimes associated with episodic and catastrophic events[20]. Previous studies have also shown that stress changes due to flank failure may enhance future volcanic activity[21,22], alter the eruptive style and the chemistry of the erupted lavas[23–26] and promote fault reactivation[27]. As a consequence of a flank collapse, some volcanoes display the reorganisation of crustal magma reservoirs[28], while other volcanoes show a migration, or the development of new volcanic rift zones[29].

Field data[12,15,30], analogue experiments[31,32] and numerical models[33] have been used to demonstrate that flank collapses may affect the orientation of dykes and the distribution of volcanic vents. Because dykes tend to be oriented perpendicular to the minimum compressive stress, they can be used to identify paleo stress fields. For instance, Tibaldi[30] inferred from dyke orientations and dip angles within the volcanic edifice of Stromboli volcano, Italy, that the stress field due to the topography of the collapse embayment is responsible for the orientation of dykes in the vicinity of the collapse scarp. McGuire and Pullen[31] used analogue models to explain the orientation of fissures at Mt. Etna considering regional and gravitational stresses due to the volcanic cone and the Valle del Bove topographies. Acocella et al.[32] showed with similar analogue experiments that dykes injected within a volcanic cone and in the vicinity of the collapse topography, would tend to reorient parallel to the collapse scarp. Furthermore, Tibaldi et al.[33] used numerical, finite-element models, to show that the stress field within the volcanic edifice associated with the topography of the collapse embayment at Stromboli volcano is able to explain the orientation of dyke intrusions parallel to the collapse scarp.

These studies addressed the mechanical effect of a lateral collapse on dyke intrusions at shallow depth, within the volcanic edifice and could explain the occurrence of volcanism in proximity of the collapse scarp. However, further observations on the distribution of volcanism after large collapses at intraplate volcanic islands, suggest that large-collapse events may inhibit the volcanic activity at the pre-collapse centre of volcanism, and that post-collapse volcanic activity often focusses at new locations within the collapse depression. This effective relocation of volcanism and focusing into the collapse embayment have been observed at numerous volcanic ocean islands[15,29,34,35], including but not limited to Fogo (Cabo Verde), El Hierro, La Palma and Tenerife (Canary Islands), St. Vincent and Martinique (Lesser Antilles), La Réunion (France) and Stromboli (Italy) (cf., Figs. 1 and 2).

In the study at hand, we design a mechanical model to reproduce and quantify the relocation of eruptive vents following a giant lateral collapse. Complementary to previous studies, we compute the post-collapse stress field within the crust, below the volcanic edifice and couple this with a model simulating the propagation of magmatic dykes through the crust.

Recent studies demonstrated that unloading forces introduced by large surface mass redistributions, as they occur, e.g., during caldera collapse or continental rift formation, have the potential to significantly change the orientation of magmatic intrusions, even at a depth of several kilometres within the crust, and therefore dictate the location of post-event volcanic activity[36–38]. While the stress field within a volcanic edifice with collapse topography has been previously calculated[33], the stress change induced by unloading forces below the volcanic edifice, and its effect on the relocation of volcanic activity following large flank collapses, has not yet been investigated. Our model also accounts for the loading stress associated with the growth of the pre-collapse volcanic edifice. By considering separate contributions for the volcanic loading and collapse unloading, we examine the loading stress dissipation on the timescale of the edifice construction (Myr), and the purely elastic stress change associated with the flank collapse.

We use a 2D boundary element (BE) model which simulates magmatic dyke propagation[39] in order to compute the most favourable pathways for magma ascent from the base of the crust to the base of a volcanic edifice that has been affected by a flank collapse. We chose to base our numerical model on the example of Fogo volcano, Cabo Verde, because the post-collapse growth of the Pico do Fogo stratocone represents a striking example of vent relocation that occurred as a consequence of the collapse of the volcano's eastern flank[36]. The simulated dyke paths are compared to the location of post-collapse volcanism at Fogo and discussed in the light of observations from several examples worldwide, particularly from ocean island volcanoes, where a large number of

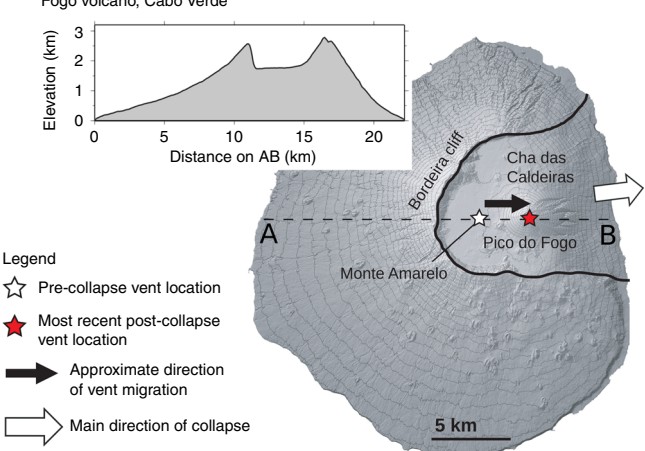

**Fig. 1** Schematic sketch of Fogo volcano. The figure shows the volcano collapse embayment (black solid line), the location of the main pre- and post-collapse centres of volcanism (white and red stars, respectively), the direction of the collapse and the direction of the shift of the volcanic activity (white and black arrows, respectively). The topography profile of Fogo along the AB segment (dashed line) is plotted as an inset

lateral collapses has been recognised[3,11,40,41], and a shift and clustering of post-collapse volcanic activity within the collapse embayment has been highlighted by several previous studies[29,34,35,41–43].

## Results

**Observations at Fogo volcano.** Fogo is located in the Atlantic Ocean off the coast of West Africa. It is the only historically active volcanic island of the Cabo Verde archipelago[44]. The island is built up from the remnants of a single, steep sided Monte Amarelo volcano that featured a central summit crater and multiple-flank vents distributed over three radial rift zones (NNE, SSE and WSW/WNW) that were centrally fed by laterally propagating dykes[35]. A catastrophic lateral collapse (with a volume of ~130–160 km³) occurred ~73 kyr ago and caused the whole eastern side of the volcano to slide into the ocean[45]. The

remaining prominent lateral collapse amphitheatre has been gradually filled by the new Chã das Caldeiras volcano with two prominent NNE and SE rift zones, a diffusely defined WSW rift zone and a centre, the Pico do Fogo cone, that is shifted ~4 km to the east with respect to the inferred centre of the pre-collapse edifice[35]. Most of the post-collapse activity and all historic eruptions took place within the collapse scar[35, 46] that is enclosed by the almost vertical Bordeira cliffs. However, during the early growth of the Chã das Caldeiras volcano, eruptive activity has persisted along the older Monte Amarelo rift zones at levels of or below the Chã das Caldeiras plain[35]. A thin layer of post-collapse lavas may even exist on the lower levels of the WSW rift zone (which is referred to as Ribeira do Pico member[35]); nevertheless, this rift zone (within and outside the Bordeira escarpment) has been the least productive since the lateral collapse of the Monte Amarelo volcano[35].

**Model constraints.** We apply our model to a vertical W-E cross section of the crust, beneath Fogo Island, intercepting the centre of the former summit of the Monte Amarelo volcano. We base this decision on the existence of preferential N-S-trending post-collapse intrusions in the Chã das Caldeiras[35], the presence of two approximately N- (NNE) and S- (SSE) oriented rift zones and the geometry of the flank collapse (W–E oriented). We consider dykes starting at the depth of the Moho, as both petrological and geodetic studies show no evidence of shallow magmatic reservoirs within the crust beneath Fogo volcano[23,47,48].

We compute the stress field within the crust beneath Fogo volcano, taking into account the loading stress due to the mass of the volcano, the unloading due to the collapse of its eastern flank and the reloading associated with the post-collapse edifice regrowth (Fig. 3a). The surface force distributions are constrained

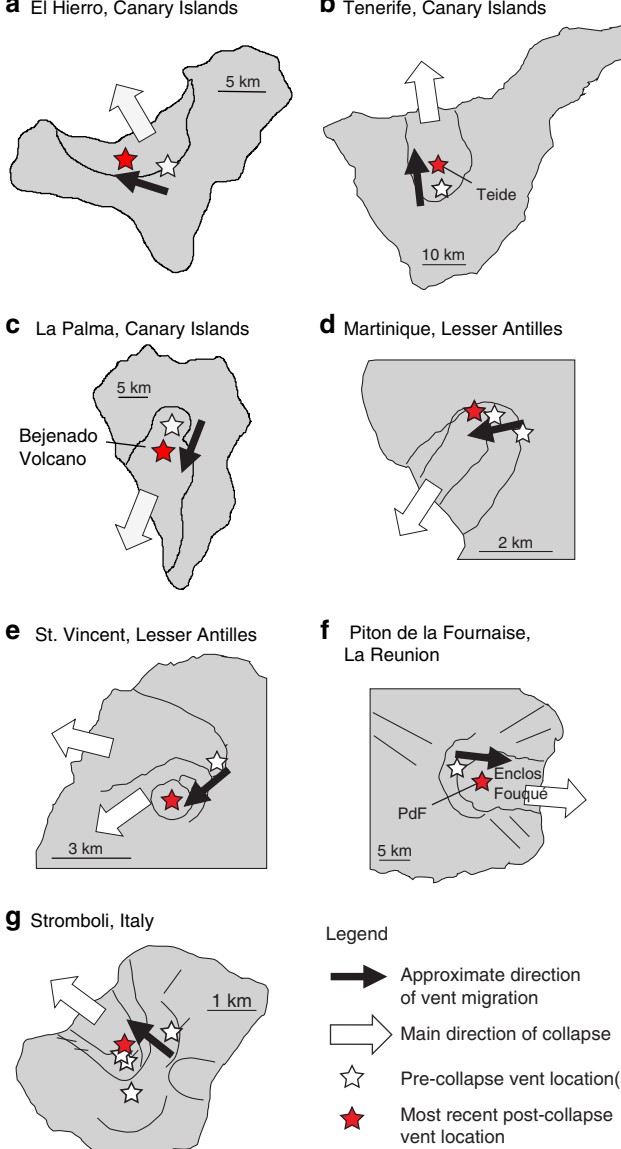

**Fig. 2** Examples of volcanic ocean islands with large collapses. **a-g** Schematic sketches of volcanic ocean islands that show evidence of one or multiple large-scale lateral flank failures and the related migration of volcanic activity (cf., Table 1 and the references therein)

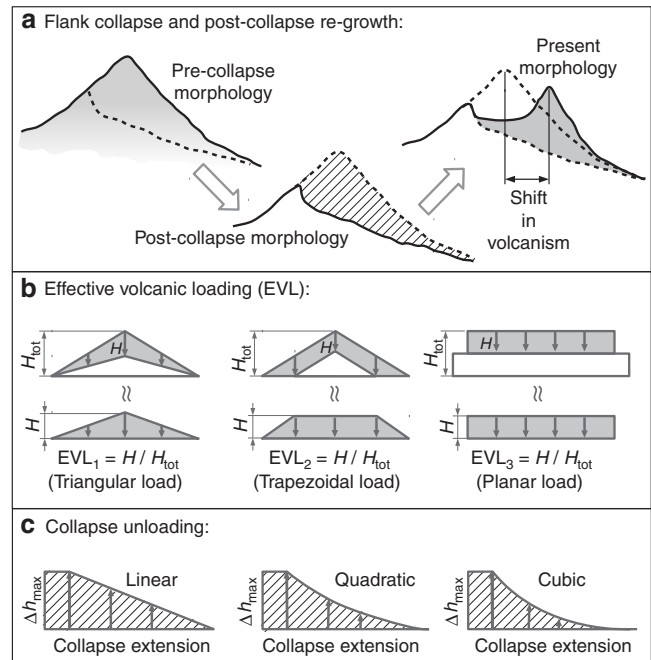

**Fig. 3** Effective volcanic loading and collapse unloading. Illustration of the basic components of our stress model: **a** morphological evolution of the profile of a volcano experiencing a flank collapse and a shift in volcanic activity, **b** effective volcanic loading due to the pre-collapse volcanic edifice. Grey areas represent the actual loading force distribution corresponding to a certain EVL type and magnitude, **c** unloading force distributions used to simulate the mass removed by the flank collapse event

by basic topographic information, as extracted from pre- and post-collapse elevation profiles[35]. We vary the geometry and the intensity of the pre-collapse volcanic loading by introducing the concept of "effective volcanic loading" (EVL, Fig. 3b), because the loading stress which has been released within the crust during the growth of the volcanic edifice cannot be easily constrained. Such stress release may occur as a result of different processes, such as previous magmatic intrusions or fluid migration, earthquakes and viscous relaxation[49]. Therefore, different EVL scenarios are applied to represent the elastic stress due to the mass of the pre-collapse volcanic edifice, which is effectively acting within the crust before the collapse. We also vary the profiles of the unloading force distribution associated with the collapse (Fig. 3c). In this way, we account for the uncertainties in the pre- and post-collapse topography profiles. The results show stress changes due to the growth and collapse of the volcanic edifice, which influence the dyke propagation paths.

**Dyke propagation paths**. Our simulated dyke trajectories tend to drive magma to both sides of the area where we impose the maximum unloading (i.e., the previous location of the Monte Amarelo summit, as indicated by the white star in Fig. 1). In other words, our model results indicate that the collapse of the eastern flank of the Monte Amarelo volcano may favour magmatism on both the eastern and western flanks of the volcano, while none of our model runs result in dyke paths that reach the base of the former Monte Amarelo summit within the Chã das Caldeiras (see Fig. 4).

Our results show that the effective volcanic loading (EVL) magnitude is the parameter that most significantly affects the position of dyke arrivals at the base of the volcano (i.e., 2000 m below the sea level) (Fig. 4a–c and Supplementary Figs. 1–3). We find that dyke trajectories are also sensitive to the shape of the load which we refer to as the EVL type (Fig. 4d–f and Supplementary Figs. 1–3), but less sensitive to the different unloading profiles that we tested (see Fig. 4g–i and Supplementary Figs. 1–3).

Furthermore, we find that the smaller the EVL magnitude, the deeper the dykes deviate from the initial vertical propagation direction, and the deeper this deviation, the further away dykes propagate from the centre of the Monte Amarelo volcano (Fig. 4a–c). The amount of effective loading needed in order to explain magmatism at the location of the Pico do Fogo stratocone is EVL, which is ~0.4–0.6 (cf., Fig. 4a–c). This means that effective loading stress acting within the crust at the time of the flank collapse was between 40 and 60% of the total elastic stress, while the other portion had been dissipated during the growth of the volcano (i.e., before the collapse, as further explained in the section "Stress model for volcanic loading", in "Methods" section).

Model results obtained using different EVL types show that a triangular loading (EVL$_1$) seems to focus the dyke trajectories more at the centre of the load. This effect is much less pronounced when using a trapezoidal load (EVL$_2$) and disappears for a planar shape of the load (EVL$_3$). The results obtained for EVL$_2$ and EVL$_3$ are indeed rather similar. This also implies that our model is only sensitive to the exact location of the pre-collapse topographic summit when EVL$_1$ is applied (cf., Fig. 4d–f and Supplementary Fig. 1).

The modelled unloading area associated with the flank collapse has an extension of 13 km and a maximum elevation difference of $\Delta h_{max} \sim 1600$ m, centred at the location of the Monte Amarelo summit. We find that the different unloading profiles that we tested do not significantly influence the dyke trajectories on the western side of the volcano. However, the dyke arrivals at the eastern flank are slightly shifted westward using concave

unloading profiles (see Fig. 4g–i). Also, from Fig. 4g–i (linear quadratic and cubic unloading profiles, respectively), dyke trajectories become more symmetrical with respect to the pre-collapse central vent ($x = -4$).

Furthermore, we find that less buoyant dykes get arrested at a deeper level underneath the western flank (in the vicinity of the collapse scarp), as compared to dykes propagating towards the eastern flank of the volcano (underneath the collapse embayment) (Fig. 4l–n and Supplementary Fig. 4). We explain this with the decompression gradient beneath the collapse scarp which favours dyke propagation along the trajectories that point to the unloaded area (i.e., the eastern flank in our Fogo example). This is in line with the results obtained by Pinel and Jaupart[21], who found that edifice destruction processes may allow more dense magma to be erupted. According to our results, the minimum volume of buoyant magma (with $\Delta\rho = 300$ km m$^{-3}$) that is needed for the dykes to reach the eastern flank of the volcano is ~$2.5 \times 10^{-3}$ km$^3$ (with a critical dyke length of ~3.2 km, cf Sect. "The dyke propagation model" in "Methods" section). Using this volume, those dykes propagating towards the western flank stall between 1.1 and 2.5 km below the base of the volcanic edifice (i.e., the elastic surface of our model, where we apply the loading and unloading forces due to the topography). The corresponding volume needed for the dykes to reach the western flank is about one order of magnitude higher: ~$3.5 \times 10^{-2}$ km$^3$ (with a critical dyke length of ~6.2 km). This result was obtained for the scenario shown in Fig. 4l.

In the last step, we tested the effect of progressive refilling of the collapse embayment. We find that the dyke trajectories focus at the location of the initial dyke arrivals as an effect of the additional load. Iterating this procedure, and therefore increasing the load on the eastern flank, has the effect that progressively more dykes are attracted towards the eastern flank. This in turn results in a small but gradual westward migration of vents at the eastern flank (see Fig. 5a–b and Supplementary Figs. 5–6). For our model parameters, we obtain a westward drift of the eastern dykes in the order of 1–2 km, though not reaching the location of the pre-collapse volcano centre.

**Effective loading and unloading and dyke volume**. From our simulated dyke trajectories, we infer that the percentage of loading stress related to the volcano's formation that was preserved in the crust at the time of the flank collapse (i.e., not released during the growth of the volcanic edifice) was between 40 and 60% (EVL = 0.4–0.6, Fig. 5c). This estimate has to be considered as the elastic stress due to loading needed to obtain a post-collapse location of volcanism, which resembles the location of Pico do Fogo (cf., dashed contour lines in Fig. 5c).

For each of the stress scenarios, we calculated the unloading vs. loading ratio, as the ratio between the total unloading forces and the total effective loading forces applied at the upper boundary of the model (cf., Fig. 5c, Supplementary Figs. 1–3 and Supplementary Table 1). The smallest unloading/loading ratios that we tested resulted in a negligible effect on the dyke propagation paths (red colour contour in Fig. 5c). From this, we infer a "critical" unloading/loading ratio which is needed in order to generate a shift in the location of volcanism. We computed such critical ratios for each combination of EVL geometry and the unloading profile (Supplementary Table 2), and found that they range between 7 and 25%. In other words, for flank collapses that removed more than 7% of the EVL, a shift of volcanism may be expected. Note that these are the ratios between the "unloading" over "effective loading" forces which are computed on a 2D cross section of the volcano, assuming a plane strain approximation, and therefore, they do not directly translate into a volcano-over

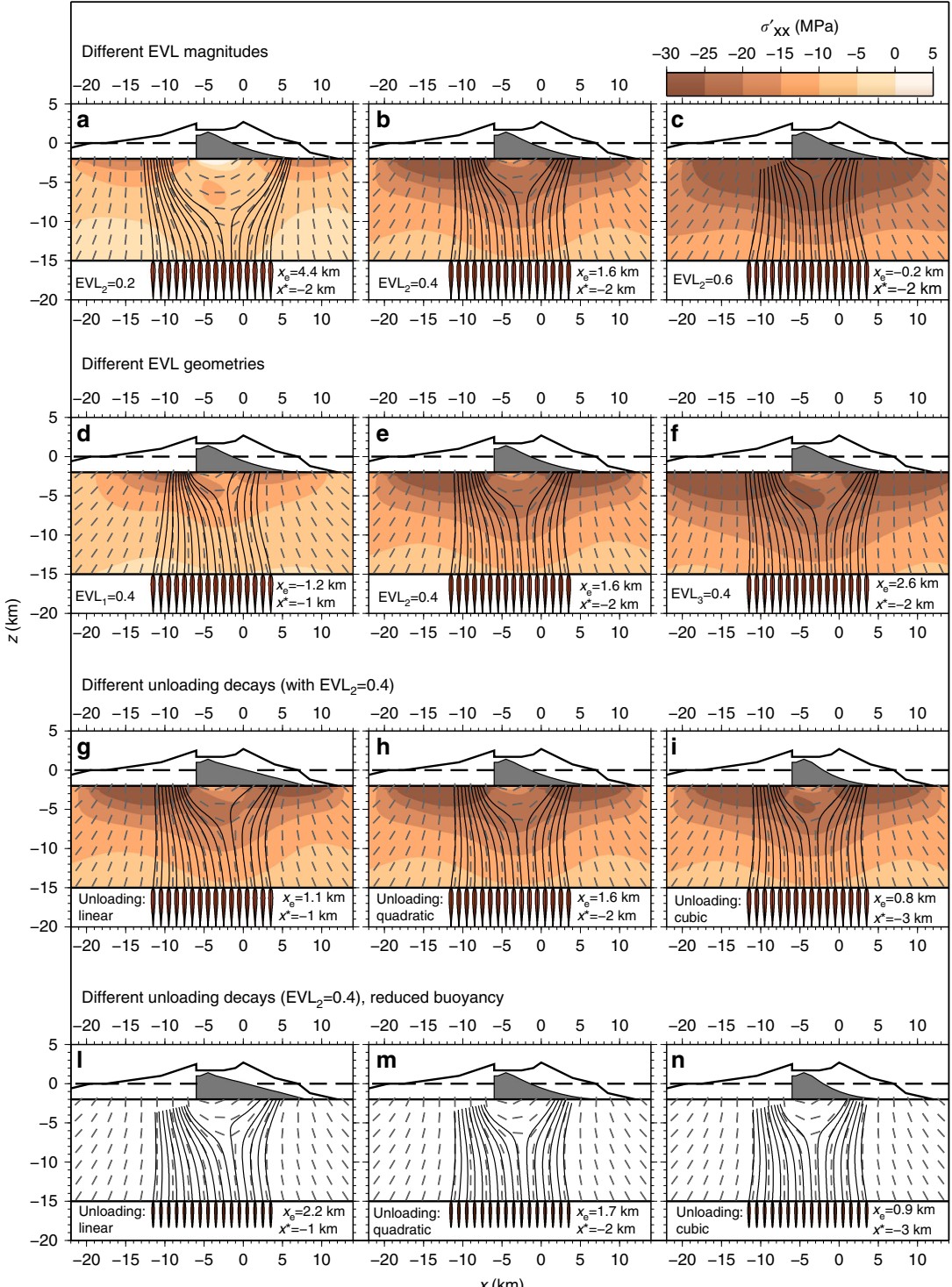

**Fig. 4** Dyke propagation paths. **a–c** Dyke trajectories with increasing effective volcanic loading (EVL), and constant unloading (quadratic decay), **d–f** dyke trajectories with constant EVL magnitude, constant unloading (quadratic decay), but with varying EVL types (d-triangular, e-trapezoidal and f-planar), **g–i** dyke trajectories with $EVL_2 = 0.4$ but with different unloading (linear, quadratic and cubic decay, from **g** to **i**, respectively), **l–n** same as **g–i**, respectively, but with reduced buoyancy of the dyke intrusions. In all panels, the colour contour is the horizontal stress change due to the topography (superposition of loading and unloading stresses). The grey dashed lines indicate the directions of maximum compression. The dyke initial cross sections are shown in red. Dyke paths are the black solid curves. The dashed line at $z = 0$ represents the sea level, and the solid line at $z = -2$ km is the base of the volcanic edifice (upper boundary of the numerical model). The grey area represents the unloading force distribution used to simulate the flank collapse. A simplified cross section of the W-E topography profile of Fogo Island is shown in each panel. $x = 0$ is the position of the Pico do Fogo stratocone. $x_e$ is the arrival position of the upper tip of the closest dyke to Pico do Fogo. $x^*$ marks the position at the base of the crust ($z = -15$ km), where dykes separate between those that will propagate towards the western flank and those that will propagate towards the eastern flank of the volcano. The initial dyke distribution is centred at $x = -4$ km, i.e., the centre of the pre-collapse volcanic cone

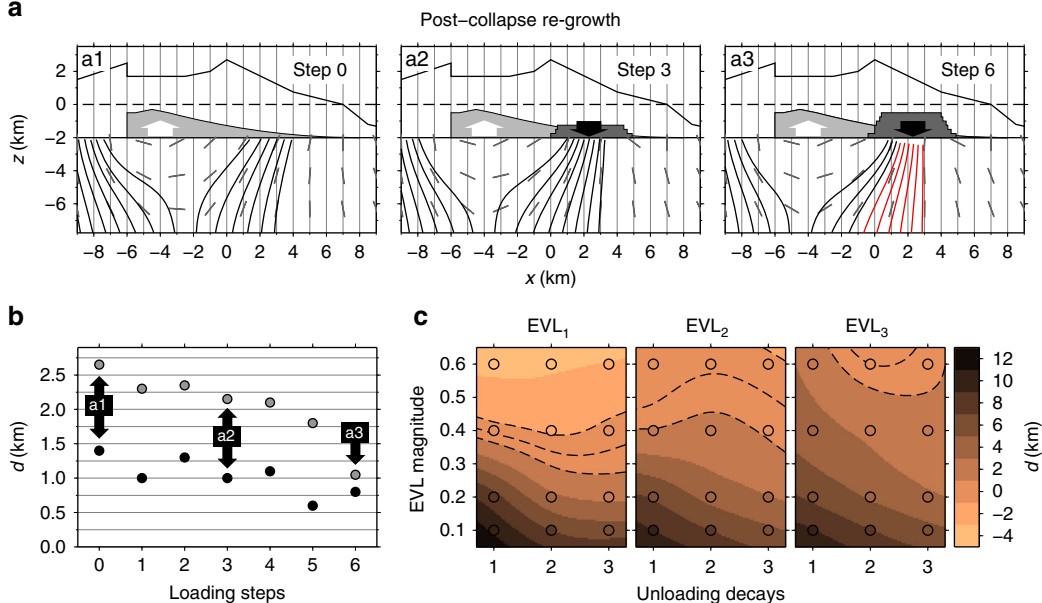

**Fig. 5** Model results for progressive refilling of the collapse scar. **a** Simulated trajectories for loading steps 0, 3 and 6, for the post-collapse regrowth scenario. Here, we used $EVL_2 = 0.4$ with quadratic unloading (scenario **b** in Fig. 4). Red paths represent dykes that cannot reach the upper boundary of the model (the base of the volcano). **b** The $y$ axis represents the eastward distance $d$ from the Pico do Fogo stratocone, while the increasing loading steps correspond to the progressive refilling of the collapse scar. Loading steps 0, 3 and 6 are shown in a1–a3. Grey dots mark the average distance $d$ from the Pico do Fogo stratocone at which eastward-propagating dykes reach the base of the volcano, and black dots correspond to the closest dyke arrivals with respect to Pico do Fogo. In **c**, the eastward distance $d$ from the Pico do Fogo stratocone at which our dyke model predicts volcanism is colour coded as a function of the EVL magnitude vs. unloading type: 1 – linear, 2 – quadratic and 3 – cubic. Each circle represents a scenario for which we simulated the propagation of 16 dyke intrusions. The dashed lines mark the contours for which $d = −1$, 0 and 1 km. The relative magnitudes of unloading vs. loading forces, for each stress scenario, are listed in Supplementary Table 1

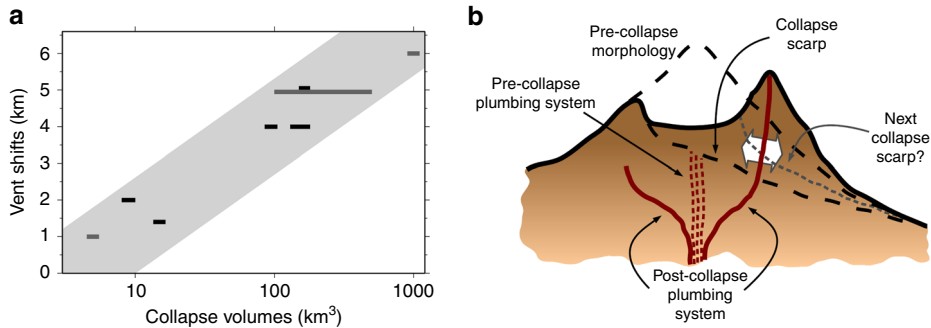

**Fig. 6** Vent shifts vs. collapse volumes and summary cartoon. **a** Shows the vent shifts (in km) between the centre of pre-collapse volcanism and the location of post-collapse volcanism as a function of the collapse volumes. All shifts occurred towards the collapsed flank. Vent shifts and collapse volumes are also listed in Table 1. We assume a 20% error for those collapse volumes for which no error estimates are available (i.e., Tenerife, La Palma, Martinique, St. Vincent and Stromboli, cf., Table 1). Black segments refer to volcanoes with no evidence of shallow reservoirs, while grey segments refer to volcanoes where shallow reservoirs exist (cf., Table 1). The sketch in **b** illustrates the modified plumbing system after the lateral collapse, as it may appear according to our model results. The dashed grey line represents the location of a potential renewed flank instability promoted by the post-collapse regrowth of the edifice and the new plumbing system

collapse-volume ratio. Nevertheless, this implies that it largely depends on the size of the lateral collapse with respect to the EVL, whether and how far dyke trajectories in the crust, beneath the volcanic edifice, are deflected (cf., Fig. 6a).

Our results for low-buoyancy dyke intrusions show that unloading-induced stress gradients favour the propagation along dyke pathways pointing towards the collapsed flank. This indicates that even though modelled dyke trajectories predict magmatism on both sides of the volcano upon a flank collapse event, magmatism on the collapsed flank is more likely. The most recent dyke intrusion at Fogo volcano (during the 2014–2015

eruption) carried a volume of magma that was estimated to be between $2.7 \times 10^{-3}$ and $3.7 \times 10^{-3}$ (km³)[48] (Supplementary Table 3). This volume compares very well to our estimate of the minimum volume of magma needed to reach the eastern flank of the volcano, which is $\sim 2.5 \times 10^{-3}$ km³ (corresponding to a dyke length of 3.2 km, an average opening of $\sim 0.25$ m and a maximum opening of $\sim 0.4$ m), and it is one order of magnitude smaller than the critical volume needed to reach the western flank. However, our estimate depends on a variety of parameters: we found that when increasing the buoyancy by 33% (from 300 to 400 kg m⁻³), the critical volume decreases by $\sim 60\%$ (from $\sim 2.5 \times 10^{-3}$ to

**Table 1 Vent shifts and collapse volumes**

| | Collapse volume | References | Presence of a shallow reservoir (depth b.s.l.) | References | Vent relocation | References |
|---|---|---|---|---|---|---|
| Fogo | Between ~130 and 160 km$^3$ | Masson et al.[51] | No (< 16 km) | Amelung and Day[23] | ~4 km | Day et al.[35] |
| El Hierro | Between ~150 and 180 km$^3$ (El Golfo) | Krastel et al.[11] | No (11–15 km) | Becerril et al.[71] | ~5 km | Carracedo et al.[34] |
| Tenerife | ~1000 km$^3$ (cumulative of 3 collapses) | Krastel et al.[11] | Yes (~2 km) | Ablay and Marti[72] | ~6 km | Carracedo and Troll[44] |
| La Palma | ~95 km$^3$ (Cumbre Nueva) | Krastel et al.[11] | No | Amelung and Day[23] | ~4 km | Carracedo et al.[34] |
| Martinique | ~(13 + 2) km$^3$ (2 collapses) | Le Friant et al.[43] | No (7–12 km) | Martel et al.[73] | ~(1 + 0.4) km | Le Friant et al.[43] |
| St. Vincent | ~ 9 km$^3$ | Le Friant et al.[42] | No (~6 km) | Feuillard et al.[74] | ~2 km | Le Friant et al.[42] |
| La Reunion (Enclos Fouque) | Between ~100 and 500 km$^3$ | Oehler et al.[41] | Yes (~0 km) | Peltier et al.[75] | ~5 km | Lénat et al.[50] |
| Stromboli | ~(2.2 + 1 + 1 + 0.7) km$^3$ (4 collapses) | Tibaldi[76] | Yes (~2 km) | Landi et al.[77] | < 1 km | Tibaldi et al.[33] |

Volumes of collapses, depth of the reservoirs and shift in post-collapse volcanism at Fogo and at the other volcanic islands shown in Fig. 2. Here, shallow reservoirs are defined as < 5 km b.s.l.

~0.95×10$^{-3}$ km$^3$). When increasing $K_r$ by 85% (from 100 to 185 MPa m$^{1/2}$), the critical volume increases by 120% (from ~2.5×10$^{-3}$ to ~5.5×10$^{-3}$ km$^3$). Variations of the critical volumes as a function of other rock and magma parameters are shown in Supplementary Table 3.

## Discussion

The results obtained for the Fogo case study explain the shift of active vents within the collapse embayment after a sector collapse, as proposed by Day et al.[36], based on morphological evidence. Our findings have important general implications for many other intraplate ocean island volcanoes. A number of examples with striking similarities in the migration of volcanic activity following a volcanic flank collapse are shown in Figs. 1 and 2 and listed in Table 1.

Among the Canary Islands, El Hierro (Fig. 2a) experienced repeated giant lateral collapses. The most spectacular and recent one coincides with the abrupt end of volcanic activity along the central rift. The post-collapse volcanic activity partially refilled the collapse embayment[24,34]. At Tenerife (Fig. 2b), the post-collapse Teide cone is shifted seawards with respect to the inferred pre-collapse morphology[43]. Similarly, at La Palma (Fig. 2c), the Bejenado volcano lays within a giant landslide amphitheatre that was created during the lateral collapse of the southern flank of the older volcano[34].

The islands Martinique and St. Vincent are among the better-known examples from the Lesser Antilles. Martinique (Fig. 1d) experienced multiple collapses at the north-western part of the island[43]. The most recent flank collapse event occurred after the growth of a cone inside an older horseshoe-shaped collapse structure. All post-collapse eruptive vents are located within or along the edge of the oldest collapse scarp[42]. At Soufrière volcano, located on St. Vincent (Fig. 2e), two horseshoe-shaped structures resulting from flank collapses exist[41]. The older of the two scarps cuts the pre-collapse volcanic edifice and hosts a post-collapse volcanic cone. This again has been partly truncated by the second collapse, upon which the scarp was partially refilled by new lavas[42].

At La Réunion (Fig. 2f), the Piton de la Fournaise volcano developed within an older, 8 km wide, caldera-like, gravitational collapse structure referred to as the Enclos Fouqué[6,40,50]. At Stromboli volcano (Fig. 2g), a shift of the centre of volcanic activity of several hundred metres occurred upon the first of four subsequent collapses of the volcano's north-west flank[33]. These four collapses appear to be nested and seem to have moved slightly towards the sea[33].

An additional example may be Santo Antao (Cabo Verde), where the Tope de Coroa volcanic peak dominates the topography of the southwestern flank of the island. Similarly to the Pico do Fogo stratocone, it lies within a horseshoe-shaped escarpment that has been interpreted as a landslide scar[51]. However, in this case, the location of pre-collapse volcanism is not well constrained.

Moreover, the relocation of volcanic activity in response to flank movements can affect a whole volcanic rift zone. One very famous example can be found on the Island of Hawaii. Here, a link exists between the westward migration of the south-west rift zone of Mauna Loa and the gravitational seaward slumping of the western flank of the volcano[29].

Even though none of such giant flank collapse events have been identified anywhere on Earth since the Quaternary[51], large-scale flank movements currently occur at Kīlauea volcano, Hawaii[20,52], Piton de la Fournaise[53], Mt. Etna[27] and possibly elsewhere in the world. Therefore, large-scale flank failure may occur in the future.

We observe that the smaller collapses in Table 1 (e.g., Martinique, St. Vincent and Stromboli) show smaller shifts between pre- and post-collapse volcanic vent locations, while larger collapse volumes (such as Fogo, El Hierro and La Palma) correspond to larger shifts of volcanism (Fig. 6a). These observations can be explained by our model results.

Some of the presented examples feature shallow magma reservoirs within or right below the volcanic edifice (i.e., Teide, Piton de la Fournaise and Stromboli). Even though the shifts in pre- and post-collapse vent locations (~6 km at Teide, Tenerife and ~5 km at Piton de la Fournaise, La Réunion) generally agree with our model results (as the collapse volumes are similar or just slightly larger, as compared to Fogo volcano), a direct comparison is rather difficult given the fact that the volume uncertainties for these two cases are very large (Table 1 and Fig. 6a). At Stromboli, the relatively small volumes of each of the multiple collapses of the NW flank fit well with the episodic migration of volcanism. For such small collapses, the effect of unloading forces below the volcanic edifice might be negligible. In fact, at Stromboli, a prominent consequence of the lateral collapses is the reorientation of dykes parallel to the collapse scarp, in the vicinity of it[15,33].

Many of the above examples have in common that flank collapses reoccur at the same locations or slightly further downslope (e.g., at Piton de la Fournaise, St. Vincent, Martinique, Stromboli and others). This has been noticed by several previous studies[9,12,13,15]. A self-sustaining mechanism has been proposed, whereby scarp-parallel dyke emplacement is promoted by the

collapse topography and flank movements, and recurrent phases of flank instability are triggered by dykes striking in that direction[9,12,15]. In addition, a feedback effect has been proposed[15], whereby the decompression resulting from a sector collapse causes the drainage of magma until a critical mass of the cone is again reached, followed by another collapse of the same flank.

The fact that in our model, dyke trajectories below the volcanic edifice are deflected towards the collapse embayment, supports the concept of a feedback effect between repeated collapses and edifice reloading[15], and provides an efficient way of reloading the collapse embayment mainly.

In the chain of consequences, the location of future collapse scarps will also be affected by a shift in the pathways of dykes (Fig. 6), as the dyke opening itself functions as a source of instability for steep volcanic flanks[17,18,52].

All volcanoes considered in this study (Table 1) are situated in a rather neutral tectonic environment. Conversely, if the magnitude of the tectonic stress is large enough, the unloading effect of the flank collapse may be masked. This prevents us from comparing the results that we obtained for Fogo with other examples such as arc volcanoes and volcanoes in rift systems. An additional major difference between ocean island and arc volcanoes is the viscosity of magma, which might also affect dyke widths and their trajectories. In fact, our dyke propagation models better apply to rather low-viscosity magmas[54]. At Mount St. Helens, for instance, the 1980 flank collapse has not been followed by a shift of the location of volcanic activity. The growth of a new lava dome occurred over the course of several episodes approximately at the same location as the pre-collapse summit of the volcano, within the collapse embayment. The volume of the Mount St. Helens collapse was estimated to ~1.5 km³; this is rather small when compared to the examples listed in Table 1 (with the exception of Stromboli). No dyke-like intrusions have ever been detected at Mount St. Helens. Another arc volcano which experienced a larger flank collapse is the Mt. Shasta volcano, Northern California, with a collapse volume of ~45 km³ which is one of the largest known in the Quaternary[5]. After the collapse, a new cone formed that was shifted ~2.5 km towards the collapse direction, within the inferred collapse embayment[55]. However, before that activity, a first cone formed only ~0.5 km from the location of the pre-collapse summit, then the vent shifted as described above and finally, the latest volcanic activity restarted at the approximate location of the previous summit[56]. Petrological studies suggested that several crustal reservoirs exist at different depths between ~4 and ~25 km beneath Mt. Shasta[56], which could imply that cones at different locations might be linked to reservoirs at different depths.

Our model, applied to Fogo Island, can explain the location of the Pico do Fogo stratocone, provided that the elastic loading due to the growth of Fogo volcano, which was actually preserved in the crust at the time of the collapse ranges between 40 and 60%. Our results have general implications for volcanic islands worldwide. We found that large lateral collapses, with an unloading of more than 7% of the EVL on the crust, may promote a significant deflection of magmatic intrusions beneath the base of the volcanic edifice towards the flanks of the volcano, favouring the collapsed embayment (Fig. 6). Moreover, by addressing the effect of volcanic flank collapses on the propagation paths of magmatic intrusions below a volcanic edifice, our study complements previous works that addressed the effect of flank collapses on the eruptive style and frequency, and on the chemistry of magmas[21–25,33,57]. Therefore, our concept contributes to an improved understanding of the mutual interactions between the structural evolution of a volcano and its eruptive activity.

## Methods

**The dyke propagation model**. Dykes in our model are represented as boundary element mixed-mode cracks in plane strain approximation. The model describes a vertical cross section of the dyke intrusion, perpendicular to the dyke plane. The dykes open under assigned normal and shear stresses that are the magma overpressure and the shear component of the topographic stress, respectively. The overpressure along the dyke is given by the difference between the magma pressure and the confining stress. The latter is defined as the superposition of the lithostatic pressure (isotropic and depth dependent) and the normal component of the topographic stress. Topographic stresses are computed using analytical formulas for loading and unloading forces at the surface of an elastic half-space[57]. Loading and unloading force distributions are used to model the loading of the pre-collapse volcanic edifice and the stress change induced by the flank collapse. The effect of loading is expected to deflect dyke trajectories towards the loaded region at the surface of the crust[39,59,60], while unloading forces tend to deflect dyke trajectories towards the sides of the unloaded region[36–38]. The dyke propagation model accounts for the magma buoyancy and compressibility, while it neglects magma viscosity. As a consequence, the magma overpressure profile within the dyke is linear along the depth and proportional to the magma-rock density difference (magmastatic overpressure profile). The dykes are nucleated at the base of the crust (i.e., at 15 km below sea level). The trajectories within the crust are determined by testing the incremental elongation of dykes in different directions. Our algorithm chooses the direction where the elastic and gravitational energy releases are maximal[39,60]. In the current application, we varied the dyke buoyancy (which is done by either changing the magma-rock density contrast or the dyke volume). This allows us to estimate the depth and location where intrusions may get stuck, particularly when the volume of the magma is close to the critical volume for propagation[61]. As the initial model input is the cross section of the intrusion (i.e., a "volume per unit length" along the out-of-plane direction), we estimate the 3D volume of the intrusion assuming that the width of the intrusion (out-of-plane dimension) is equal to the dyke length[59]. Our estimate of a critical volume for propagation depends on various parameters such as rock fracture toughness ($K_r$), magma buoyancy ($\Delta\rho$), bulk modulus of magma ($K$) and rock rigidity ($G$). Among these parameters, the most important ones are $K_r$ and $\Delta\rho$ to which the critical volume is quite sensitive (cf., Supplementary Table 3). In the study at hand, we used a magma-rock density contrast $\Delta\rho$ ranging between 200 and 400 kg m$^{-3}$ (with magma densities $\rho_m = [2300; 2500]$ kg m$^{-3}$) and a bulk modulus for magma $K = [10; 50]$ GPa (according to ref. [62]). We set the shear modulus of the rock to $G = 20$ GPa, Poisson's ratio to $\nu = 0.25$, the rock fracture toughness to $K_r = 100$ MPa m$^{1/2}$ (according to ref. [54]) and the density of the crust to $\rho_c = 2700$ kg m$^{-3}$ (according to ref. [62]).

**The stress model for volcanic loading**. The pre-collapse volcanic edifice is an ~40-km-wide volcano of an ~5000-m elevation (above the ocean floor), which is centred at the location of the summit of the Monte Amarelo volcano[35]. A fundamental parameter of our model is the amount of elastic loading which is effectively acting on the crust. We introduced the concept of the EVL in order to account for the fact that during the growth of Fogo volcano (which took between 3 and 6 Myr, according to ref. [63]), the elastic loading stress induced in the crust may have been partially released by different processes, such as fluid intrusions, earthquakes and viscoelastic effects[49]. The magnitude of the EVL indicates the percentage of the elastic loading stress that is preserved in the crust, while the remainder (i.e., the total elastic loading minus the effective loading) has been released by the above-mentioned processes. Since it is difficult to estimate the amount of loading stress that was acting within the crust at the time of the flank collapse, we tested different effective loading geometries (EVL types) and intensities (EVL magnitudes). For instance, if we assume that 40% of the elastic loading stress was released during the development of the volcanic edifice, we scale the elastic loading by a factor of 0.6, and say that we use an EVL = 0.6. The different EVL geometries that were tested are a triangular load (EVL$_1$), a trapezoidal load (EVL$_2$) and a planar load (EVL$_3$), as shown in Fig. 3b. The first, EVL$_1$, represents a volcano that grows in height while at the same time maintaining a constant base area. The second was obtained by subtracting a triangular load from a larger one with identical slopes. This would be the 2D representation of a volcanic edifice that maintains a constant slope while growing, so that the effective load is shaped like a layer covering the flanks of the volcano. Finally, the third, simple planar-loading scenario was introduced in order to evaluate the effect of the edifice's slope on the dyke trajectories beneath the volcano.

**The stress model for flank collapse and post-collapse regrowth**. We model the stress changes within the crust that were induced by the Monte Amarelo flank collapse by means of unloading forces acting at the base of the volcanic edifice (i.e., at the ocean floor, ~2000 m below sea level). The unloading force distribution represents the mass removed by the flank collapse, which is the difference ($\Delta h$) between the W-E-trending topography profiles before and after the collapse (see Fig. 3a, c), multiplied by the density of shallow rocks ($\rho_s = 2000$ kg m$^{-3}$) and the acceleration due to gravity ($g = 9.81$ m s$^{-2}$). The maximum amount of unloading, the corresponding location and the total extension of the unloading area are fairly well constrained by the results of previous studies[35,51]. The amount of maximum unloading is constrained by the maximum height difference between the Monte

Amarelo volcano and the Chã das Caldeiras plain ($\Delta h_{max} \sim 1600$ m). It is centred at the former Monte Amarelo summit and extends over an area of ~2 km (cf., the pre-collapse Monte Amarelo volcano profile given in Fig. 9 in ref. [35]). The region of unloading extends to ~13 km eastward from the Bordeira escarpment. The eastern boundary of the unloading area is constrained by the submarine sudden change in slope where the landslide debris starts being exposed (cf., Fig. 4 in ref. [51]). Since we lack detailed information on the W-E topography profile immediately after the flank collapse (before the formation of the Pico do Fogo stratocone), we consider three different unloading profiles (i.e., the distribution of mass removed by the flank collapse), namely linear, quadratic and cubic decays (Fig. 3c). Finally, we tested the effect of the progressive refilling of the collapse scar on the dyke trajectories by applying a load at the location of dyke arrival. We iterate this process until the cumulative additional load compensates the maximum unloading ($\rho_s g \Delta h_{max}$).

**Limits of the model and its applicability.** One major limitation of our model is a direct consequence of the plane strain approximation on which the boundary element (BE) code for dyke propagation relies. The 2D nature of our model restricts our results to a cross section of the 3D natural case. For the Fogo case study, this means that our results apply to N-S-trending intrusions (i.e., intrusions striking perpendicular to the model cross section, which is W-E). However, we expect our calculations to be stable for a relatively wide range of orientations (W-E $\pm$ ~20°), given the similarity of all radial topography profiles that are oriented about $\pm 20°$ with respect to the W-E direction. This holds under the negligence of the Pico do Fogo stratocone, which does not affect our stress calculation as it developed after the giant flank collapse event. Also, our plane strain approximation requires the dyke width (i.e,. the off-plane dyke extension) to be smaller than the extension of the topographic features in the same direction. We also argue that this condition is met, since the two most recent dyke intrusions (which occurred in 2014–2015 and 1995) were about 2- to 4-km long[23,48], while the whole volcanic edifice has a diameter of ~40 km and the east-facing summit collapse structure extends over ~9 km[35]. The plane strain solutions that we used to compute the effect of loading and unloading forces[59] imply unbounded displacements away from the surface load. This limitation can be overcome by considering relative displacements at the surface, which are always finite[59]. However, in our application, we do not consider the displacements induced by the loading, since we only look at the stress induced within the crust.

The formulation that we use to compute crustal stresses implies that we do not address the stress field due to the collapse within the volcanic edifice, which has been modelled by previous studies[33]. Here, we aim to investigate the effect of the stress change induced by the flank collapse within the crust, below the volcanic edifice. We neglect the shear tractions associated with the growth of the volcanic edifice (and their partial removal due to a flank collapse), which might be transmitted at the interface between the volcanic edifice and the ocean floor[17]. As shown by analytical and numerical solutions, the principal stress orientations are generally affected by this shear stress[17,64–66]. It is noteworthy that the stress field below the volcanic edifice is less affected by the approximation of vertical forces than the stress field within the edifice. In fact, numerical simulations[17] show that the stress field within the volcanic edifice is affected by the slope of the edifice, while at the base of the edifice, where our simulations stop, the directions of compressive stresses[17] are in general agreement with the orientation of the principal stresses obtained by vertical forces associated with a triangular load (EVL$_1$, see Fig. 4d, below the western side of the volcano). In contrast, for the trapezoidal- and rectangular-loading geometries (EVL$_2$ and EVL$_3$), the maximum compression vectors are more vertical (i.e., less inward dipping) than the ones obtained by Dieterich[17]. In addition, the study of Van Wyk de Vries and Matela[65] shows that stress vectors at the base of an edifice are vertical only when the substratum has a low viscosity. However, in the spatial region below the volcanic cone (where we compute the dyke paths), the directions of maximum compression are mostly vertical[65], which are in agreement with our simulations for the EVL$_2$ and EVL$_3$ scenarios (i.e., the trapezoidal and planar loadings). McTigue and Mei[64] conclude that the main differences between the vertical load approximation and a stress field which considers both vertical forces and tractions are mainly confined to shallow depths (within the volcanic edifice), and at the sides of the volcanic edifice, i.e., in regions where we do not compute dyke propagation paths.

The analytical approach that we used for the calculation of the stress beneath the volcano does not account for the mechanical interaction of the collapse with a magmatic reservoir. If the reservoir is deep enough (deeper or at the level of the lateral extension of the collapse), such interaction may be negligible. However, the mechanical interaction of a shallow reservoir (located within or right below the volcanic edifice) with a flank collapse[24,67], as well as the thermal effect due to the presence of the reservoir[68], might result in a different stress pattern with respect to the one that we considered in our study. Our results would only be relevant for those cases where the shallow reservoir developed at a later stage, after the collapse. In such cases, we would speculate that the formation of a post-collapse shallow reservoir beneath the collapsed flank might be a consequence of the stress redistribution due to the collapse: magmatic dykes that are deflected towards the collapse embayment, at the base of the volcanic edifice, may intrude at this location, and progress towards a shallower depth (i.e., within the volcanic edifice) also favoured by unbuttressing stress. Within the volcanic edifice, magmatic dykes

may stall and accumulate because of low-density layers and horizontal discontinuities[69], or because of the stress induced by possible flank movements which would foster increased intruded vs. erupted magma volumes[57].

In our stress model, surface loading and unloading forces act on an elastic half-space. Therefore, our model does not account for the bending of the lithosphere. The effect of lithosphere bending beneath the loading and close to its edges causes further horizontal compression near the surface and extension at the base of the lithosphere. The opposite (shallow extension and deep compression) is expected on the sides of the loaded region, where the bending is convex[49,70]. The magnitude of bending stresses depends on the location at which the effect of loading is considered. When we compute dyke trajectories, in a region centred beneath the volcano, the effect of bending should be less significant than at the edges of the load[70]. In addition, lateral variations of bending stresses have characteristic wavelengths of ~100–200 km and therefore should not introduce significant lateral stress gradients within the region that we considered for dyke propagation (~15–20 km).

In conclusion, our simplified method to compute the loading stresses, and in particular the plane strain approximation imposes a number of limitations to the applicability of our models. The advantage of such simplifications, however, is that we are able to introduce the concept of EVL and that we can explore the effect of the elastic stress release which may occur during the growth of the volcanic edifice. In addition, by varying our simplified loading geometries (EVL types), we are able to explore the effect of different orientations of the principal stresses close to the loaded area. Most importantly, the plane strain approximation allows us to simulate the dyke propagation paths for the different stress scenarios. As of the time of this writing, no 3D models exist that have the capability to simulate the propagation of magmatic dykes and at the same time allow for complex trajectories interacting with a heterogeneous external stress field (as our 2D model for dyke propagation does). Our dyke propagation model accounts for many physical parameters characterising the crust and the intrusions, such as rock rigidity, Poisson's ratio, fracture toughness, magma buoyancy and dyke overpressure. This facilitates a novel quantitative comparison between the dyke deflection and the magnitude of the stress change due to the collapse.

An extension of the applicability of our model to more complex scenarios, which may include, for instance, the mechanical and thermal effect due to a shallow reservoir, possible strong heterogeneities in the crust or interactions with tectonic forces and 3D topography, may be achieved in the future by coupling our dyke propagation model with the stress field provided by dedicated, e.g., finite-element models. This would allow us to reduce some of the current assumptions and simplifications.

**Limitations in data and observables.** In our model, we varied the geometry and the intensity of the EVL, the geometry of the collapse segment and the buoyancy of the magma. In this way, we were able to discriminate the most critical parameters that affect the trajectories of magmatic dykes in our simulations. The results show that the location of the post-collapse eruptive activity highly depends on the interplay between the amount of the EVL and the unloading due to the collapse, and therefore on the difference between the pre- and post-collapse topographies. Especially, the pre-collapse topography may only be vaguely reconstructed. Moreover, it is difficult to quantify the amount of stress which has been released within the crust over long time scales (Myr), by brittle rock failure, fluid intrusions and other non-elastic processes. We introduced the flexible parameter EVL in order to compensate for this lack of knowledge.

The volume of the 2014–2015 dyke intrusion at Fogo volcano, as estimated by means of InSAR measurements[48] results in the volume accommodated by the dyke close to the surface, approximately between 0- and 2-km depth. Similarly, our estimate for the critical volume is given for a compressible magma at vanishing confining stresses; therefore, the two can be directly compared. However, we estimate the (3D) critical volume from a plane strain dyke model (2D cross section), based on the assumption of a constant dyke width (out-of-plane length) that is equal to the dyke length. This might systematically overestimate the 3D volume, as in reality, we expect the tip of a 3D hydrofracture to be rounded in the direction of propagation[59]. On the other hand, we are also neglecting the magma loss in the dyke tale due to cooling and viscous effects, and therefore, the critical cross section for propagation computed by our model may be underestimated. In addition, several other parameters (e.g., the rock fracture toughness and the magma buoyancy) would also affect the critical volume for propagation (cf., Supplementary Table 3). The limitations rising from poor constraints in model parameters are generally difficult to overcome and result in uncertainties in our simulations. A way to address uncertainties in such cases is to make use of a probabilistic approach. For instance, assigning likelihood distributions to the input parameters, running the model several (thousands of) times and computing the probability to get a certain result (i.e., a "family" of dyke propagation paths). However, stochastic approaches are computationally expansive and would require further optimisation of our code and parallelisation of the model runs, which for the here-applied model needs to be addressed in the future.

**Data availability.** All the relevant data that have been used in the present study are available from the authors.

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

## Acknowledgements

We thank Mehdi Nikkhoo and Eleonora Rivalta for their valuable help in discussing the modelling approach and its applicability. We also thank Marco Bagnardi and Falk Amelung for their helpful discussions about the most recent intrusions at Fogo volcano. This study is a contribution to VOLCAPSE and EPOS projects, which has been funded by the European Research Council under the European Union's H2020 Programme, grant numbers [ERC-CoG 646858] and [676564], respectively.

## Author contributions

F.M. and T.W. planned the paper and the numerical simulations. F.M. implemented and ran the B.E. code. N.R. provided geological constraints, as well as the geomorphological data and interpretations for the numerical simulations. All authors contributed in comparing the numerical simulations with the Fogo case study. All authors contributed to Figs. 1 and 2. F.M. created Figs. 3–6 and Table 1, with input from N.R. and T.W. All authors discussed the results and their general implications, and contributed in writing the manuscript, led by F.M.

## Additional information

**Competing interests:** The authors declare no competing financial interests.

