## [Peer Review File · Nature Communications]

Reviewers' comments:

Reviewer #1 (Remarks to the Author):

This paper investigates, by numerical modelling, the effect of huge lateral collapse of a volcano on the distribution of stresses in the underlying crust. The simulation has been done based on bibliographic data of Fogo Volcano (Cabo Verde). Based on the difference of the reconstructed stress field between the pre-collapse and the post-collapse settings, the authors suggest possible magma migration along diverted dyke paths. The main result is that the magma pathways after a huge lateral collapse tend to cluster below the collapse favouring eruptive vents there.

Although the argument of the relationships between lateral failure of volcanoes and reorganization of magmatic centers is interesting, this paper has a series of serious issues that do not make it suitable for the high standard of NATURE Communications. The suggestion of rejection is based on the lack of novelty, on the insufficient conceptual advance, and on the technical approach. Here below I pinpoint these major problems.

First of all, the basic idea and general results of this paper have essentially already been put forward by several previous authors. The change in the stress field below a volcano sector collapse has already been published based on numerical modelling. Similarly, the focusing of volcanic activity within a sector collapse has already been suggested. Reorientation of dykes along the scarps of sector collapses has already been demonstrated based on field data, on numerical modelling and even by physical modelling. These are just some of the papers dealing with that: McGuire and Pullen, 1989; McGuire et al., 1990; Carracedo et al., 1999; Day et al., 1999; Tibaldi, 2003, 2004; Walter and Troll, 2003; Acocella et al., 2005.

Another "new" suggestion claimed by this paper "By revealing the tendency of magmatic activity to refill the collapsed embayment with new lavas, we provide a mechanism that may also explain the cyclical, repeated failure of the same volcanic flank, and the consequent asymmetrical growth of the edifice observed at many volcanic islands", in reality is similar, or identical, to previous suggestions contained in several papers also referred to onshore volcanoes, among which I recall: Siebert, 1984; Siebert et al., 1987; McGuire et al., 1990; Beget and Kienle, 1992; Elsworth and Voight, 1995; Voight and Elsworth, 1997; Voight, 2000; Donnadiu et al., 2001; Tibaldi, 2001; Hildenbrand et al., 2004.

A conceptual advance might be obtained at least by proposing an improvement in the technical approach, but this paper failed on this since a 2D computation approach was used; since the problem of mass distribution and its changes is related to the failure of part of conical edifices, a 3D approach would have been more appropriate and new.

The chapter "Implications" in reality is more appropriate to be included in the Discussion section. In the "Method" section, the authors suggest that a shallow magma chamber should form below the volcano collapsed sector, but it is not clear why and how it should form if unbuttressing, due to the sector failure, should instead facilitate magma upwelling.

A pdf with suggestions is attached.

Resuming, if improved this paper might be of interest for other journals such as JVGR, but it does not contain important advances of significance to specialists to be published in Nature Communications.

Reviewer #2 (Remarks to the Author):

The paper develops model for dike propagation in volcanoes, previously loaded by the edifice load, and subjected to unloading by a giant landslide, and then reloaded by a new growth. The paper shows some interesting results that provide an explanation for the shift in eruptive centers after giant lateral collapses at volcanoes. The general result obtained makes it of interest for a broad community. The dike propagation models are sound. However, I have some concerns on the method used to compute stresses related to loading and unloading of the volcano by vertical

forces only, disregarding the shear coupling with the substratum. Would the same results be obtained if the actual forces corresponding to the volcanic edifice loading and unloading were considered? The paper also misses a quantification of the relative effects and amplitudes of the loading and unloading, taken separately and together. The paper would be of a broader general interest if there was a discussion on volcanoes where this shift is not observed. Is there a reason why this shift is only observed at oceanic islands? An other major issue concerns the research quoted. It is quite frequent that there is a misunderstanding on what a paper actually showed. Some major references are also forgotten. In order for the paper to be published in nature communications, the authors should address these weak aspects of their paper.

Major concerns:

My main concern is about the way the edifice load is applied. Indeed, the edifice load is applied as a series of vertical forces only, and the shear stresses induced in the elastic medium by this load are completely neglected. Indeed, because rocks are elastic and not fluid, shear stresses are transmitted because each new lava flow is solid and mechanically coupled to the substratum. For instance the numerical solutions of Dieterich (JGR, 1988, Figure 5) shows that, at the base of an edifice, stress vectors are centripetal and not vertical. Similarly, the study of van Wyk de Vries and Matela (JVGR, 1998, Fig. 4) shows that stress vectors at the base of an edifice are vertical only when the substratum has a low viscosity. I should also say that it is an usual way to proceed when investigating the effect of loads and unloads (Pinel et al., Philosophical Transactions of the Royal Society, 2000; JVGR, 2010; Albino et al., GJI, 2010), but it does not mean it is right to do so. This effect should be discussed, and the difference between the stress field obtained taking into account the actual loading by a topography and the vertical force approximation should be discussed as it may strongly affect the results. To conduct this discussion the authors could rely on previous publications. Additionally to both studies previously mentioned, there are some asymmetrical analytic solutions for volcanoes derived using perturbation methods (For instance McTigue and Mei, International Journal for Numerical and Analytical Methods in Geomechanics, vol 11, 1987; or Pan et al., Int. Jour. Rock. Mech. Mining. Sci., 1995), and some 2D plane and axisymmetrical numerical solutions (Cayol and Cornet, JGR, 1998, Currenti and Williams, Phys. Earth Planetet. Int., 2014; Ernst et al.,). Some more elaborated numerical solutions have been calculated by Cianetti et al. (GJI, 191, 2012) and chaput et al. (GRL, 2014) in order to take into account the initial stress field related to an edifice loaded by gravity. In these articles, the authors incrementally apply the gravitational load making sure that they do not induced displacements. Finally McGovern and Solomon, JGR, 1998 consider the state of stress related to an incrementally loaded edifice. Similarly as for the loading, it is desirable to discuss the influence of this assumption on the unloading and later reloading which again neglect shear stress changes.

An other concern is that the joint effect of the edifice load over unloading is presented throughout the study, with no discussion on the respective effects of the edifice load and unloading taken separately. Throughout the text, there is no mention of the relative magnitude of the loading over the unloading stresses although I figure that this ratio plays a major role in the propagation paths described. Because of this, Figure 3a and lines 137-139 present a counter-intuitive results, with the smallest edifice loads inducing larger path change for the dike, whereas it is probably the large relative magnitude of the withdrawn edifice volume over the initial volume which creates this result. In this respect it is puzzling when the authors mention a loading/unloading ratio (l 182), as this concept has not been introduced. I must say I do not understand how it is computed. A value of 6% is given, but this ratio seems to be an unloading/loading ratio rather than the contrary (loading/unloading).

An other concerns is that results are only presented for volcanoes where there is a shift in vent location after a major collapse. What about Mount St Helens, for instance? This volcano is cited in the introduction, and the paper does not attempt to explain the eruption pattern at this volcano with their model. It seems the volcano for which the model works are all ocean island. How could this be explained?

It is indicated that and EVL (Effective Volcano Loading) of 0.4 indicates that the effective loading stress acting in the crust represents 40% of the total elastic stress. It would be useful to explain where the rest of the elastic stress come from (gravity loading of the half-space, perhaps ?) and what is the representative volume considered, as this amount will depend on the considered volume.

Articles are not always cited appropriately, and the referencing is too often imprecise: sometimes the results mentioned do not support the author argument. For instance Cayol et al., 2014 (l. 41) did not really show that dykes affect flank stability, better references would be Iverson, *JVGR*, 1995 or Chaput et al., *GRL*, 2014. Similarly, Cayol et al., 2000 (l. 45) did not show that flank failure enhances volcanic activity. It is rather the contrary. Flank failure increased intruded versus erupted magma volumes; but the paper showed that dike intrusions interact with decollement faults through stresses and may lead to catastrophic flank failure. Neither Cayol et al., 2014 nor Delaney and Denlinger (l. 262) showed that dyke opening was a source of instability for steep volcanoes.

The article sometimes cites authors who cited other author. The initial article should be mentioned. Anderson (1989) and Turcotte and Schubert (2014) are not a handbook with material properties. Note that the laboratory densities of small samples might very well be different from the in situ densities at larger scales. There are many such examples.

The references are not always up to date. For instance the collapse volume at Piton de la Fournaise given in Table 1 was reevaluated by Oehler et al., *Bull. Volc.*, 2004, 2008. Similarly, this reference would be a better one to document the Enclos Fouqué. An other good reference for large scale flank motions at Hawaii is Ando (*JGR*, 1979) which documented the 1975 M7.2 EQ. For Piton de la Fournaise, better references than Cayol et al., 2014, (l. 241) for flank motions are Froger et al., *JVGR*, 2015 and Tridon et al., *JGR*, 2016.

Some important references are omitted. Syracuse (*JGR*, 2010) is an other good reference for interactions leading to catastrophic failure (l. 241). Pinel and Jaupart (*Philosophical Transactions of the Royal Society*, 2000) showed how failure may affect chemistry of the emitted lavas. This reference could be given in the introduction (l. 47) and further discussed when investigating vent shifts as a function of magma chemistry.

Minor concerns:

The results section should only contain results and not methods. The two first paragraphs concern the method: morphology choice, stress model. They should not appear in a (new) result section.

The supplementary materials are very useful. They are rarely mentioned in the text. Only S1 is mentioned. They should be mentioned along with the results

The authors (line 114) mention stress release. Which is the mechanism of such a stress release ? Do the authors mention relaxation processes ?

It would be interesting to see a plot showing the relative volume of unloading over edifice loading versus the amount of vent shift for various edifices in the world, ie ocean volcanic island, intraplate volcanoes, basaltic, andesitic and dacitic volcanoes, , may be as an introduction. That would make the paper of a more general interest.

Figures :

Figure 1 It seems some of the information displayed in the figures refer to Table 1. It should be indicated.

Figure 3. Refers to x_c as the magma arrival position in the caption, but this position is referred to as x_e in the figure. It would help to indicate the relative magnitude of the unloading relative to ELV.

Figure 4. ELV magnitude as a function of the unloading decay is presented. Again, what is the

relative ratio of the loading/Unloading ?

Other comments:

In the discussion, it would also be interesting to relate the edifice destruction, to the capacity of the volcano to emit more dense magma as indicated by Table 1. This result confirms a previous study by Pinel and Jaupart (Philosophical Transactions of the Royal Society, 2000), it should be mentioned.

Valerie Cayol

Reviewer #3 (Remarks to the Author):

Review for Nature Communication manuscript NCOMMS-16-29851, "The effect of giant lateral collapses on magma pathways and the location of volcanism", by F. Maccaferri et al.

In this paper, existing field observations of eruptive vent shift in location following major sector collapses are explained by a 2D numerical model developed by the author regarding dyke path in non-homogeneous stress field. By combining the 2, the authors have a way to explain, for the first time, the shifting eruptive centers following major mass redistribution (e.g. sector collapse, giant landslide) at ocean island volcanoes.

This paper presents compelling evidence and would greatly contribute to readers from the field, as well as from other fields as they might get inspired to develop this approach for their own field, or to contribute on the non-elastic part of the problem.

The author will see that I have one main concern (detailed below) that could be easily fixed by making the discussion more general, and leaving aside the details about actual volume involved.

I certainly hope that the authors will find those comments sound and useful.

Sincerely,

Benoit Taisne, Earth Observatory of Singapore

Comments regarding volumes:

2D vs 3D

Throughout the paper volume extracted from 2D modeling are given in km^3 , and compared, as they are, with volume extracted from deformation data. This is misleading, and might result in an apparent perfect match between the minimum volume needed for eruption, and volume estimated using InSAR data (in the case of Fogo volcano).

Using volume and critical length line 163, thickness in 2D is 80cm, while using the ones given line 167, 2D thickness is 5.6m, while adding the third dimension, say, 100m to 1000m (as suggested Table s1, Gonzalez et al. 2015), this thickness drops to less than a centimeter... making it quite challenging to migrate at all (thermal cooling will be dominant at this stage).

Unless I missed the information about the third dimension this part should be changed or removed (without impeding the quality of the paper).

Minimum volume needed

In the same spirit as the previous point between 2D and 3D volume, I don't think this add much value to the paper. A more general discussion could be made on the fact that more volume/buoyancy is needed to reach the surface on one side of the volcano than for the other side. Stating that smaller volume is needed to reach the surface where indeed new eruptive activity is observed is probably enough. To strengthen the discussion, I would move lines 406 to 416 from "limitations in data and observables", into the main body of the paper where critical volumes are discussed (lines 297 to 303).

Few minor comments:

Regarding Fogo magmatic plumbing system, reference based on petrology would be useful. Constraining such a deep source using geodetic data only is not a strong argument (with respect to the size of the island itself...), at best you can probably rule-out shallow depth. I found a PhD dissertation, and abstract discussing the origin depth of the erupted magma, by Hildner Elliot, that is in accordance with the one given in the text (reference to be added line 106-107). They have a paper on the 1951 pre-eruptive source and storage that would be relevant for this study. Last line of there abstract: "Thus petrologic data indicate that flank collapse events may significantly influence deep-seated magma plumbing systems beneath ocean islands." Hildner et al., 2012, Barometry of lavas from the 1951 eruption of Fogo, Cape Verde Islands: Implications for historic and prehistoric magma plumbing systems, JVGR.
<http://dx.doi.org/10.1016/j.jvolgeores.2011.12.014>

I'm surprise by the comment line 417-422 on the possibility to include uncertainties to the results. How long does a run last? Once again, if the discussion on the volume remains qualitative and not quantitative this is not an issue anymore.

Typos/Edits:

Line 181 (reference to figure 4c): I would suggest adding a dashed line instead of mentioning the orange colour contour.

Line 353: Boundary Elements instead of BE (line 68 is far at this point)

Figure 1 would gain by adding the 2D cross section considered for Fogo volcano, as well as adding the rift zone for Fogo.

Figure 3: x_c in the caption refer to x_e in the figure.

Point-by-point response to the reviewers.

Manuscript: “The effect of giant lateral collapses on magma pathways and the location of volcanism”.

Reviewer#1: This paper investigates, by numerical modelling, the effect of huge lateral collapse of a volcano on the distribution of stresses in the underlying crust. The simulation has been done based on bibliographic data of Fogo Volcano (Cabo Verde). Based on the difference of the reconstructed stress field between the pre-collapse and the post-collapse settings, the authors suggest possible magma migration along diverted dyke paths. The main result is that the magma pathways after a huge lateral collapse tend to cluster below the collapse favouring eruptive vents there.

Although the argument of the relationships between lateral failure of volcanoes and reorganization of magmatic centers is interesting, this paper has a series of serious issues that do not make it suitable for the high standard of NATURE Communications. The suggestion of rejection is based on the lack of novelty, on the insufficient conceptual advance, and on the technical approach. Here below I pinpoint these major problems.

Reply: We appreciate the comments and improved the manuscript accordingly. These comments helped us in particular to better highlight the novelty of our research and the conceptual advances achieved by applying our modelling technique. We followed point-by-point the specific comments of all the reviewers, and while revising the manuscript, we paid particular attention to the general and constructive criticism raised by reviewer#1.

Here we provide a revised version of the manuscript wherein we better distinguish between earlier works and already published ideas and our specific achievements (see answer to point 1, reviewer#1). We also improved the description of the concepts that were already developed by previous authors and improved the revision of existing literature (see answer to point 1, reviewer#1 and point 5, reviewer#2). In the final discussion we recall those previous concepts and better highlight how they are supported and further strengthened by our results (see answer to point 2, reviewer#1). We are confident that by addressing these major concerns raised by reviewer#1, we could greatly improve the manuscript, better highlight the novelty of our achievements, and better clarify how they integrate within the larger picture and state-of-the-art research in the field.

1) First of all, the basic idea and general results of this paper have essentially already been put forward by several previous authors. The change in the stress field below a volcano sector collapse has already been published based on numerical modelling. Similarly, the focusing of volcanic activity within a sector collapse has already been suggested. Reorientation of dykes along the scarps of sector collapses has already been demonstrated based on field data, on numerical modelling and even by physical modelling. These are just some of the papers dealing with that: McGuire and Pullen, 1989; McGuire et al., 1990; Carracedo et al., 1999; Day et al., 1999; Tibaldi, 2003, 2004; Walter and Troll, 2003; Acocella et al., 2005.

Reply: Here the reviewer raises three major issues (we will refer to them as 1.1, 1.2 and 1.3), the first being that:

1.1) “[...] the basic idea and general results of this paper have essentially already been put forward by several previous authors. [...] Similarly, the focusing of volcanic activity within a sector collapse has already been suggested. [...]”

Reply: A shift and focusing of the post-collapse magmatic activity within the collapse embayment

has already been observed at several volcanoes. This idea was developed and put forward by observational studies, such as Corracedo et al., 1999 (El Hierro and Las Palma), Day et al., 1999 (Fogo), and others (cited in our manuscript in Tab. 1). Even though the phenomena has already been observed and described, we provide for the first time a mechanical model that might actually explain and quantify the shift of post-collapse volcanism within the collapse embayment (previous mechanical models addressing the effect of a collapse on magmatic dykes could explain their orientation and distribution along the collapse scarp, c.f. *Tibaldi et al., 2008*).

Earlier studies and the observations therein represent our motivation to develop a model that couples the stress change induced by the collapse event, and the (effective) loading due to the pre-collapse edifice, with simulations of the propagation of magmatic dykes, from Moho depths to the base of the volcano.

We thank the reviewer for his/her comment, and we now clearly distinguish between observations and ideas that have been put forward by previous studies and highlight that our model present a mechanism which might explain and quantify the observed shift and focusing of volcanic activity within the collapse embayment for the first time (lines **46-48 / 76-78 / 81-86 / 90-93 / 278-280** in the revised manuscript with highlighted changes, and revised abstract).

The second major issue raised by reviewer#1 in his/her first comment concerns the fact that:

1.2) “[...] The change in the stress field below a volcano sector collapse has already been published based on numerical modelling. [...]”

Reply: Tibaldi et al. (2008) used a numerical, finite element model to compute the stress field due to a sector collapse topography. However, Tibaldi et al., (2008) and our model are complementary, and consequently explain distinct observations (as further explained in the next reply to reviewer#1 comment). While the upper boundary of our model domain is the ocean floor, i.e. the base of the volcanic edifice, Tibaldi et al. (2008) looked at the stress distribution within the edifice, at much shallower depths. We are therefore confident that our results, coupled with a dyke propagation model, complement and further strengthen the ground for Tibaldi et al., (2008), rather than duplicating their stress model. In fact, by showing that magma would preferentially enter the volcanic edifice below the collapse embayment, our model “feeds into” the modelling domain of Tibaldi et al., (2008), providing the dykes that possibly reorient later on in response to the local stress field acting within the volcanic edifice (depending on their overpressure relative to the magnitude of the deviatoric stresses in the edifice, as shown for instance by Watanabe et. al., 2000 and Dahm, 2000). This important clarification has been added in the revised version of the manuscript at lines **72-75 / 90-93 / 99-107** (please refer to the version with highlighted changes), we thank the reviewer for pointing it out.

We further notice that the techniques used to compute the stress field – within the volcanic edifice, in Tibaldi et al., 2008 – and – in the crust below the volcano, in our study – are different, and are both appropriate to investigate the effect of a sector collapse at different depths, in complementary space domains. Tibaldi et al. (2008) looked at the stress field associated to the topography of a collapse embayment, considering a gravitationally loaded volcano with collapse topography. In our study we consider loading and unloading force distributions to simulate the effect of: i) the growth of the volcanic edifice and ii) the flank collapse, and the associated stress change within the crust. In addition, we take into consideration that edifice growth and collapse act on very different time scales: the ‘Effective Volcanic Loading’ account for the stress released during the slow process of volcanic edifice growth, in contrast with the purely elastic stress change due to the instantaneous collapse event.

The third concern of reviewer#1 in his/her first comment is:

1.3) “[...] Reorientation of dykes along the scarps of sector collapses has already been

demonstrated based on field data, on numerical modelling and even by physical modelling. These are just some of the papers dealing with that: McGuire and Pullen, 1989; McGuire et al., 1990; Carracedo et al., 1999; Day et al., 1999; Tibaldi, 2003, 2004; Walter and Troll, 2003; Acocella et al., 2005.”

Reply: We agree. However, the reorientation of dykes along the scarps of sector collapses is actually not the issue we aim to address with our study. Instead, we focus on the shift of volcanic activity from the pre-collapse summit of the volcano to a new location within the collapse embayment, that goes along with the growth of a new volcanic cone. We thank the reviewer for pointing this out. We clarified this in the revised manuscript (please refer to lines **63-81 / 336-339** in the manuscript with highlighted changes).

Among the studies cited here by the reviewer, Carracedo et al., (1999) and Day et al., (1999), addressed the issue we aim to model (the shift of volcanic activity, and its clustering within the collapse embayment) based on field data. Therefore they represent, together with the other studies cited in our manuscript, the main motivation for our work. The other studies mentioned here addressed the reorientation of dykes along the collapse scarp by means of field data (McGuire and Pullen, 1989; McGuire 1990; Tibaldi, 2003 and 2004), analogue experiments (McGuire and Pullen, 1989; Walter and Troll, 2003; Acocella et al., 2005) and numerical models (Tibaldi et al., 2008, not explicitly mentioned, but most likely considered by the reviewer, here). Noteworthy, at Fogo, there are no observations of dyke intrusions parallel to the collapse scarp, showing that the two effects (the reorientation of dykes close to the collapse scarp, and the clustering of post-collapse volcanic activity within the collapse embayment), even though they are both related with the collapse event, are distinct. In fact, all the models that successfully explained the dyke orientations parallel to the collapse scarp, considered the shallow stress field within the volcanic edifice (also the analogue experiments were performed injecting fluids directly within the gelatine cone). Our model explains for the first time the clustering of activity within the collapse embayment by looking at the deflection of dyke paths at depth, below the volcanic edifice. Basically, the stresses within the volcanic edifice considered by Tibaldi et al., (2008) and in the analogue experiments of McGuire and Pullen (1989), Walter and Troll (2003), and Acocella et al. (2005), explain the distribution and orientation of fissures and shallow dykes, parallel to – and in the vicinity of - the collapse scarp. The stress change induced by the collapse within the crust, below the volcanic edifice, addressed by our study, shows for the first time a deflection of dyke pathways beneath the volcanic edifice, which would drive magmatic dykes preferentially towards (and below) the collapsed flank of the volcano, promoting magmatic activity in there.

These distinctions were not explained in the first version of our manuscript. We thank the reviewer for pointing out this fundamental point. We added these arguments in the revised version of the manuscript (please refer to lines **63 to 81** in the manuscript with highlighted changes).

2) Another “new” suggestion claimed by this paper “By revealing the tendency of magmatic activity to refill the collapsed embayment with new lavas, we provide a mechanism that may also explain the cyclical, repeated failure of the same volcanic flank, and the consequent asymmetrical growth of the edifice observed at many volcanic islands”, in reality is similar, or identical, to previous suggestions contained in several papers also referred to onshore volcanoes, among which I recall: Siebert, 1984; Siebert et al., 1987; McGuire et al., 1990; Beget and Kienle, 1992; Elsworth and Voight, 1995; Voight and Elsworth, 1997; Voight, 2000; Donnadieu et al., 2001; Tibaldi, 2001; Hildenbrand et al., 2004.

Reply: We deleted the sentence cited by the reviewer (lines **392-395**) as we agree it was misleading. The message we aimed to deliver was that our modelling results support the already existing idea that multiple collapses are encouraged to occur at the same location. We wanted to stress that our results link well with the concept of multiple collapses, and once more provide further solid ground for previous studies that investigated this problem. We thank the reviewer for rising this important

issue, we modified the manuscript according to his/her comment (please refer to lines 39-42 / 342-356 in the manuscript with highlighted changes).

Furthermore, the reviewer's comment convinced us about the need of revising the introduction and final discussion. In the revised manuscript, we carve out where our study adds a new piece of information and to which extent our results support and complement previous studies.

3) A conceptual advance might be obtained at least by proposing an improvement in the technical approach, but this paper failed on this since a 2D computation approach was used; since the problem of mass distribution and its changes is related to the failure of part of conical edifices, a 3D approach would have been more appropriate and new.

Reply: It is true that a 3D approach would represent a further technical improvement, nevertheless important implications can be drawn from our 2D model. With this respect, we note that the effect of 3D topography is less important for some situations, as also testified by previous several previous studies, such as for instance Cayol et al. (1999), in Science, who used a 2D approach to address the interaction between flank instability and shallow dyke emplacement at Kilauea volcano. In the revised version, we improved the discussion of the limitations of our model, with particular attention to the 2D vs 3D modelling issues. Maccaferri et al. (2014), in Nat. Geo., used plain strain formulation to model the unloading stress associated with the formation of a continental rift to explain the occurrence of off-rift volcanism. In the revised manuscript we better highlight the limitations and the geometrical constraints imposed by the (2D) plain strain approximation we used, as well as the advantages of it (please refer to lines 514-518 / 571-585 in the manuscript with highlighted changes).

We believe that our modelling approach do represent a “conceptual advance”: i) Introducing the Effective Volcanic Loading (EVL), our model provide a simple way to address the variability of the stress state of the crust below the volcanic edifice: The EVL is not linked to a physics-based model of the stress evolution through time before the collapse, however, by varying this parameter on a wide range of values, we could show the importance that the pre-collapse state of stress within the crust may have on the establishment of the post-collapse plumbing system. ii) In our study, we simulate the dyke propagation paths with a method that has never been applied to this problem before. As of the time of writing, no 3D models exist that simulate the propagation of magmatic dykes and at the same time allow for complex trajectories in interaction with a heterogeneous external stress field (as our 2D model for dyke propagation does). Our model for dyke propagation considers physical parameters characterising the crust and the intrusions, such as rock rigidity, Poisson's ratio, fracture toughness, magma buoyancy, and dyke overpressure. This results in a quantitative comparison between the dyke deflection and the magnitude of the stress change due to the collapse which therefore is intrinsic in our modelling technique.

Therefore, we think that an “improvement in the technical approach” is achieved by our study by coupling a plain-strain stress model with a dyke propagation model. This produces original results that in our opinion represent an advancement in the understanding of the relation between edifice collapses and the location of post-collapse magmatic intrusions.

Finally, we agree with the reviewer that a more complex stress model coupled with our dyke propagation model, would represent a further advancement. In fact, we are currently improving our model to accept stress scenarios as derived from e.g. finite element techniques as an input (revised manuscript with highlighted changes, lines 586-591).

4) The chapter “Implications” in reality is more appropriate to be included in the Discussion section.

Reply: We modified the section “**Implication**”, that is now entitled “**Significance of the effective loading, unloading, and dyke volume**”, and contains results from our model other than dyke trajectories.

5) In the “Method” section, the authors suggest that a shallow magma chamber should form below the volcano collapsed sector, but it is not clear why and how it should form if unbuttressing, due to the sector failure, should instead facilitate magma upwelling.

Reply: We appreciate this comment. Actually, our model suggests that magma would more easily intrude into the volcanic edifice below the collapsed flank (this, in fact, is also new with respect to earlier studies that explained concentric dyke orientations close to the collapse scarp). Possible magma chamber development within the volcanic edifice cannot be directly addressed by our simulations. However, we think that inside the volcanic edifice the conditions for magma to stall at shallow depth, and therefore the development of a shallow reservoir, would be possible or likely, especially when multiple intrusions accumulate, even though unbuttressing stresses would facilitate magma upwelling. This idea is based on the likely presence, within the volcanic edifice, of low density layers of less-consolidated rocks and horizontal discontinuities that may favour dyke arrest (Gudmundsson and Brenner, 2001, for instance). In addition, Cayol et al. (2000), showed that at Kilauea Volcano, Hawai’i, flank failure caused the ratio between intruded and erupted magma to increase. We agree that unbuttressing stresses may favour the upward propagation of intrusions (Tibaldi, 2004), however, this does not necessarily imply that the conditions to form a shallow reservoir cannot be matched. We included this discussion in the revised version of the manuscript (please refer to lines 551-561 in the manuscript with highlighted changes).

6) A pdf with suggestions is attached.

Reply: all suggestions in the attached pdf have been either addressed by previous answers and/or included in the manuscript.

Point-by-point response to the reviewers.

Manuscript: “The effect of giant lateral collapses on magma pathways and the location of volcanism”.

Reviewer#2 (Valery Cayol): The paper develops model for dike propagation in volcanoes, previously loaded by the edifice load, and subjected to unloading by a giant landslide, and then reloaded by a new growth. The paper shows some interesting results that provide an explanation for the shift in eruptive centers after giant lateral collapses at volcanoes. The general result obtained makes it of interest for a broad community. The dike propagation models are sound. However, I have some concerns on the method used to compute stresses related to loading and unloading of the volcano by vertical forces only, disregarding the shear coupling with the substratum. Would the same results be obtained if the actual forces corresponding to the volcanic edifice loading and unloading were considered? The paper also misses a quantification of the relative effects and amplitudes of the loading and unloading, taken separately and together. The paper would be of a broader general interest if there was a discussion on volcanoes where this shift is not observed. Is there a reason why this shift is only observed at oceanic islands? Another major issue concerns the research quoted. It is quite frequent that there is a misunderstanding on what a paper actually showed. Some major references are also forgotten. In order for the paper to be published in nature communications, the authors should address these weak aspects of their paper.

Reply: We thank Valery Cayol for her constructive comments that resulted in substantial improvements of our manuscript. We particularly appreciated the suggestion of discussing examples where a shift in the location of post-collapse volcanism has not been observed. By adding the “counter-example” of Mount St. Helens, we had the chance to broaden our discussion, and better clarify to which extent our general model can be applied to different volcanoes in different settings. Also, we very much appreciated the effort done by Valery Cayol to spot missing and/or wrong citations, giving us the opportunity to correct them.

Major concerns:

1) My main concern is about the way the edifice load is applied. Indeed, the edifice load is applied as a series of vertical forces only, and the shear stresses induced in the elastic medium by this load are completely neglected. Indeed, because rocks are elastic and not fluid, shear stresses are transmitted because each new lava flow is solid and mechanically coupled to the substratum. For instance the numerical solutions of Dieterich (JGR, 1988, Figure 5) shows that, at the base of an edifice, stress vectors are centripetal and not vertical. Similarly, the study of van Wyk de Vries and Matela (JVGR, 1998, Fig. 4) shows that stress vectors at the base of an edifice are vertical only when the substratum has a low viscosity. I should also say that it is an usual way to proceed when investigating the effect of loads and unloads (Pinel et al., Philosophical Transactions of the Royal Society, 2000; JVGR, 2010; Albino et al., GJI, 2010), but it does not mean it is right to do so. This effect should be discussed, and the difference between the stress field obtained taking into account the actual loading by a topography and the vertical force approximation should be discussed as it may strongly affect the results. To conduct this discussion the authors could rely on previous publications. Additionally to both studies previously mentioned, there are some asymmetrical analytic solutions for volcanoes derived using perturbation methods (For instance McTigue and Mei, International Journal for Numerical and Analytical Methods in Geomechanics, vol 11, 1987; or Pan et al., Int. Jour. Rock. Mech. Mining. Sci., 1995), and some 2D plane and axisymmetrical numerical solutions (Cayol and Cornet, JGR, 1998, Currenti and Williams, Phys. Earth Planetet. Int., 2014; Ernst et al.,). Some more elaborated numerical solutions have been calculated by

Cianetti et al. (GJI, 191, 2012) and chaput et al. (GRL, 2014) in order to take into account the initial stress field related to an edifice loaded by gravity. In these articles, the authors incrementally apply the gravitational load making sure that they do not induced displacements. Finally McGovern and Solomon, JGR, 1998 consider the state of stress related to an incrementally loaded edifice. Similarly as for the loading, it is desirable to discuss the influence of this assumption on the unloading and later reloading which again neglect shear stress changes.

Reply: We fully agree with Valery Cayol that a more sophisticated stress model may also consider the effect of tractions due to the coupling of the volcanic edifice and the ocean floor. As Valery Cayol suggested, we added a detailed discussion about the effect of loading and unloading when considered as vertical forces only (neglecting tractions) to the revised version of the manuscript (please refer to lines **518-541** in the manuscript with highlighted changes).

Please also find a detailed discussion below, in direct reply to Valery Cayol's comment:

One of the main limitations introduced by our simplified method to compute loading (and unloading) stresses (i.e. vertical forces), is that we can't consider the dyke propagation within the volcanic edifice, as better clarified in the revised manuscript, lines **514-518** (version with highlighted changes). However, with the current study, we aim to investigate the effect of the stress change induced by the flank collapse within the crust, below the volcanic edifice, while the stress field within the volcanic edifice has been addressed by previous authors. Noteworthy, the stress field due to the loading/unloading forces below the volcanic edifice is less affected by the approximation of vertical forces than the stress field inside the edifice. In fact, as pointed out by Valery Cayol, the numerical simulations from Dieterich (1988), Fig. 5, show that the stress field within the volcanic edifice is affected by the slope of the edifice, however, at the base of the edifice, where our simulations stop, the direction of compressive stress (as shown in Dieterich, 1988, Fig. 5), are in good agreement with the orientation of the principal stresses obtained by vertical triangular load, close to the loaded surface (Maccaferri et al., 2011, Fig. 5; Dahm, 2000). This is also replicated by our models with triangular loading (EVL1, see Fig 3b1, below the west side of the volcano), while for trapezoidal and rectangular loading geometries (EVLs 2 and 3) the maximum compression vectors are more vertical (less inward dipping), this variability is meant to represent different possible assumptions on the history of the volcanic edifice growth.

In addition, as Valery Cayol pointed out, the study of Van Wyk de Vries and Matela (1998), shows in their Fig. 4 that stress vectors at the base of an edifice are vertical only when the substratum has a low viscosity. This model is surely more complex than the one we used, however looking at the region where we focus our analysis (where the dyke paths are computed), which is a region below the volcanic cone (we do not look beyond the volcano's flanks), the maximum compression directions shown in Fig. 4 in Van Wyk de Vries and Matela (1998) are mostly vertical, just as in our simulations with EVL2 and 3 (trapezoidal and planar loadings).

McTigue and Mei (1987), conclude that the main differences between the vertical load approximation and a stress field which considers both vertical forces and tractions are mainly confined to rather shallow depths (within the volcanic edifice), and on the sides of the volcanic edifice, where we do not compute of dyke propagation paths.

In conclusion, by varying the loading geometries (EVL types) we explore their effect on different orientations of the principal stresses close to the loaded area. The triangular loading (EVL1), because of the gradient of vertical forces, results in a centripetal direction for the maximum compression (similarly to what we would have obtained considering tractions acting on the loaded region), while the planar loading geometry (EVL3) produces a rather vertically oriented σ_1 . However, tractions acting directly at the boundary between the surface and the loading may further enhance the effect which we obtain here by considering different vertical force gradients (triangular, trapezoidal and planar loading).

2) An other concern is that the joint effect of the edifice load over unloading is presented throughout the study, with no discussion on the respective effects of the edifice load and unloading taken separately. Throughout the text, there is no mention of the relative magnitude of the loading over the unloading stresses although I figure that this ratio plays a major role in the propagation paths described. Because of this, Figure 3a and lines 137-139 present a counter-intuitive results, with the smallest edifice loads inducing larger path change for the dike, whereas it is probably the large relative magnitude of the withdrawn edifice volume over the initial volume which creates this result. In this respect it is puzzling when the authors mention a loading/unloading ratio (l 182) , as this concept has not been introduced. I must say I do not understand how it is computed. A value of 6% is given, but this ratio seems to be an unloading/loading ratio rather than the contrary (loading/unloading).

Reply: Indeed, the ratio between unloading and loading forces plays a major role for the propagation paths of the simulated dyke trajectories. In the revised version of the manuscript, we describe loading and unloading separately and highlight the importance of their relative magnitudes (added at lines 255-258, manuscript with highlighted changes). The result presented in Fig. 3a is justified by the fact that the loading tend to focus dykes toward the center of the volcano and the unloading tend to deflect them to the sides: therefore, given a fixed amount of unloading a lower loading (smaller EVL) will result in a greater deflection of the dyke paths.

We improved our discussion of this relationship between loading and unloading. We also modified the text in order to introduce the concept of loading/unloading ratio and we clarified how these values have been computed (please refer to lines 242-244 / 247-249 / 415-420 in the manuscript with highlighted changes).

3) An other concerns is that results are only presented for volcanoes where there is a shift in vent location after a major collapse. What about Mount St Helens, for instance? This volcano is cited in the introduction, and the paper does not attempt to explain the eruption pattern at this volcano with their model. It seems the volcano for which the model works are all ocean island. How could this be explained?

Reply: We thank Valery Cayol for her comment. We added a discussion about Mount St. Helens and arc volcanoes in general to the final section of the revised manuscript (lines 369-385, manuscript with highlighted changes). Our model has been designed to work for ocean islands, we justify this because of the large number of observations describing the shift and clustering of volcanic activity within the collapse embayment, following a flank collapse event (please refer to lines 117-122 in the manuscript with highlighted changes). Ocean island volcanoes, as well as Fogo, are situated in a rather neutral tectonic environment. Conversely, if the magnitude of the tectonic stress is large enough, the unloading effect of the flank collapse may be masked. This prevents us from applying our model to other possible examples such as arc volcanoes and volcanoes in rift systems (please refer to lines 363-366 in the manuscript with highlighted changes). An additional major difference between ocean island and arc volcanoes is the viscosity of magma. This might play a role, since high viscosity magma may require higher overpressure to propagate. In fact, as shown by previous studies (Watanabe et al., 2000) a higher dyke overpressure decreases the dyke sensibility to external stresses, this would in turn reduce the efficiency of the mechanism presented in our study. Revised manuscript with highlighted changes, lines 366-369.

4) It is indicated that an EVL (Effective Volcano Loading) of 0.4 indicates that the effective loading stress acting in the crust represents 40% of the total elastic stress. It would be useful to explain where the rest of the elastic stress come from (gravity loading of the half-space, perhaps?) and what is the representative volume considered, as this amount will depend on the considered volume.

Reply: We introduced the concept of Effective Volcanic Loading in order to account for the fact that

during the growth of a volcanic edifice, that in the case of Fogo took ~3 to 6 Myr, the elastic loading stress induced in the crust must have been partially released by different processes such as visco-elastic effects, fluid intrusions and earthquakes, for instance. The magnitude of the EVL indicates the percent of elastic stress, due to topographic loads, that is preserved in the crust. Therefore the rest of it (i.e. the total elastic loading minus the effective loading) is assumed to be vanished, released by different processes as the ones mentioned above. In the revised manuscript we better clarified this concept, line 448-467 and also 104-107 / 195-196 / 387-388 (manuscript with highlighted changes). We thank Valery Cayol for her comment.

5) Articles are not always cited appropriately, and the referencing is too often imprecise: sometimes the results mentioned do not support the author argument. For instance Cayol et al., 2014 (l. 41) did not really show that dykes affect flank stability, better references would be Iverson, JVGR, 1995 or Chaput et al., GRL, 2014. Similarly, Cayol et al., 2000 (l. 45) did not show that flank failure enhances volcanic activity. It is rather the contrary. Flank failure increased intruded versus erupted magma volumes; but the paper showed that dike intrusions interact with decollement faults through stresses and may lead to catastrophic flank failure. Neither Cayol et al., 2014 nor Delaney and Denlinger (l. 262) showed that dyke opening was a source of instability for steep volcanoes. The article sometimes cites authors who cited other author. The initial article should be mentioned. Anderson (1989) and Turcotte and Schubert (2014) are not a handbook with material properties. Note that the laboratory densities of small samples might very well be different from the in situ densities at larger scales. There are many such examples. The references are not always up to date. For instance the collapse volume at Piton de la Fournaise given in Table 1 was reevaluated by Oehler et al., Bull. Volc., 2004, 2008. Similarly, this reference would be a better one to document the Enclos Fouqué. An other good reference for large scale flank motions at Hawaii is Ando (JGR, 1979) which documented the 1975 M7.2 EQ. For Piton de la Fournaise, better references than Cayol et al., 2014, (l. 241) for flank motions are Froger et al., JVGR, 2015 and Tridon et al., JGR, 2016. Some important references are omitted. Syracuse (JGR, 2010) is an other good reference for interactions leading to catastrophic failure (l. 241). Pinel and Jaupart (Philosophical Transactions of the Royal Society, 2000) showed how failure may affect chemistry of the emitted lavas. This reference could be given in the introduction (l. 47) and further discussed when investigating vent shifts as a function of magma chemistry.

Reply: Accepted and changes made. We reconsidered and carefully checked all citations given and made appropriate changes. In specific, we replaced the Cayol et al. (2014) reference by the work from Iverson (1995). We rephrased the main findings from Cayol et al. (2000), now stating that “the stress induced by possible flank movements [...] would foster increased intruded versus erupted magma volumes”, line 560-561. Dyke opening as a source of instability for steep volcanoes are not shown in Cayols work nor in Delaneys work, but in Elsworth and Voight (1996), for instance. We also carefully reconsidered those citations where authors cited other authors. The Turcotte and Schubert reference, however, we feel is appropriate, as a text book is considered as efficient way to establish a common knowledge base and to reduce necessary citations. We moreover consider the citation Oehler et al., (2008) for the description of the Piton de la Fournaise collapse volume, use Froger et al. (2015) for assessing its flank motion, and describe the hawaiian flank motions with reference to Ando (1979). We very much appreciate these suggestions that helped us to improve the manuscript.

Minor concerns:

1) The results section should only contain results and not methods. The two first paragraphs concern the method: morphology choice, stress model. They should not appear in a (new) result section.

Reply: The reviewer here points out what was actually also our concern, when structuring the manuscript according to Nat. Comm. standards. We thought that the structure: "introduction – results – discussion – method", with the sections “results” and “method” being the only ones that can be divided in sub-sections, necessarily implies that the section “results” has to be considered in a wide sense, and it may contain all the information that are essential in order to explain findings, including observations and important model constraints. Moving the first two paragraphs of the section “results” to the section “method” (which is the last section) would make the “results” section difficult to understand, therefore we rejected this option. The alternative would be to move these paragraphs to the end of the introduction, but in our opinion they do not belong to that section either (in the introduction they could not have a sub-paragraph title). For these reasons we decided to keep the same structure also in the revised manuscript, and renamed the first two sub-sections of the section result as “**Observations: the Fogo setting**” and “**Model constraints**”, to make clear to a reader that the section “results” includes observations and model constraints which are essential in order to understand the findings of our the study. However we are open to move these two sub-sections to the introduction if the reviewer and/or the editor do not agree with our interpretation of the structure of Nat. Comm. articles.

2) The supplementary materials are very useful. They are rarely mentioned in the text. Only S1 is mentioned. They should be mentioned along with the results

Reply: We thank Valery for this comment and added references pointing towards the supplementary material where appropriate (lines **184-187 / 202 / 213 / 231 / 244 / 248 / 275** in the manuscript with highlighted changes).

3) The authors (line 114) mention stress release. Which is the mechanism of such a stress release ? Do the authors mention relaxation processes?

Reply: The mechanism that we suggest for the release of elastic (loading) stresses in the crust are: i) viscous relaxation, which in volcanic environment may act non-uniformly in space and time, depending on the evolution of the volcanic edifice and of volcanism. ii) brittle processes such as earthquakes, slow slip events. iii) Previous dyke intrusions. The effect of all this processes tend to progressively reduce the deviatoric stress, but they cannot be easily quantified. For this reason we introduce the EVL parameters, which can be varied in order to consider different scenarios with different amount of elastic stresses preserved (or released) in the crust. We added this explanation to the revised version of the manuscript, lines **448-455** and also **163-165** (please refer to the manuscript with highlighted changes).

4) It would be interesting to see a plot showing the relative volume of unloading over edifice loading versus the amount of vent shift for various edifices in the world, ie ocean volcanic island, intraplate volcanoes, basaltic, andesitic and dacitic volcanoes, may be as an introduction. That would make the paper of a more general interest.

Reply: As we clarified in reply to the third “major concern” that was raised by Valery Cayol, we think that the results obtained for Fogo should not be directly compared with examples other than ocean island volcanoes. Nevertheless, what Valery Cayol proposes here is very interesting, and it may actually represent the base for a continuation of this study. We added in **Fig. 5 (panel a)** a plot based on the information collected in Tab. 1.

Figures:

Figure 1) It seems some of the information displayed in the figures refer to Table 1. It should be indicated.

Reply: We added a reference to Table 1 in Figure 1 caption.

Figure 3) Refers to x_c as the magma arrival position in the caption, but this position is referred to as x_e in the figure. It would help to indicate the relative magnitude of the unloading relative to ELV.

Reply: We added Table S1 in the supplementary material, showing all the unloading/loading ratios used in our simulations. We referred to Table S1 in the main text at lines **184-187**, and in **Figure 4 caption**.

Figure 4) ELV magnitude as a function of the unloading decay is presented. Again, what is the relative ratio of the loading/Unloading ?

Reply: We added a reference to Table S1 in **Figure 4 caption**.

Other comments: *In the discussion, it would also be interesting to relate the edifice destruction, to the capacity of the volcano to emit more dense magma as indicated by Table 1. This result confirms a previous study by Pinel and Jaupart (Philosophical Transactions of the Royal Society, 2000), it should be mentionned.*

Reply: We added at line **216-217** (manuscript with highlighted changes) the sentence: “[...] **in line with results obtained by Pinel and Joupert (2005), who found that edifice destruction processes may allow more dense magma to be erupted**”.

Point-by-point response to the reviewers.

Manuscript: “The effect of giant lateral collapses on magma pathways and the location of volcanism”.

Reviewer#3 (Benoit Taisne): In this paper, existing field observations of eruptive vent shift in location following major sector collapses are explain by a 2D numerical model developed by the author regarding dyke path in non-homogeneous stress field. By combining the 2, the authors have a way to explain, for the first time, the shifting eruptive centers following major mass redistribution (e.g. sector collapse, giant landslide) at ocean island volcanoes.

This paper present compelling evidences and would greatly contribute to reader from the field, as well as from other field as they might get inspired to develop this approach for there own field, or to contribute on the non-elastic part of the problem.

The author will see that I have one main concern (detailed below) that could be easily fixed by making the discussion more general, and leaving aside the details about actual volume involved. I certainly hope that the authors will find those comments sound and useful.

Reply: We thank Benoit Taisne for his very useful and constructive comments. In particular, thanks to his comment on the 2D vs 3D volume estimation, we realised that a fundamental information concerning the assumption which is at the base of our volume calculation, was missing. Also his suggestion of adding references from petrological studies on Fogo, which support one of our model assumptions (absence of shallow reservoir), was very much appreciated, and strengthen further our modelling results.

Comments regarding volumes: 2D vs 3D. Throughout the paper volume extracted from 2D modeling are given in km³, and compared, as they are, with volume extracted from deformation data. This is misleading, and might results in an apparent perfect match between the minimum volume needed for eruption, and volume estimated using InSAR data (in the case of Fogo volcano). Using volume and critical length line 163, thickness in 2D is 80cm, while using the ones given line 167, 2D thickness is 5.6m, while adding the third dimension, say, 100m to 1000m (as suggested Table s1, Gonzalez et al. 2015), this thickness drop to less than a centimeter... making it quite challenging to migrate at all (thermal cooling will be dominant at this stage). Unless I missed the information about the third dimension this part should be changed or removed (without impeding the quality of the paper).

Reply: This is a very good point and we thank Benoit Taisne for rising this issue. As Benoit Taisne mention in the last sentence of his comment, an important information on the assumption made on the third dimension of the dyke is actually missing. In fact, in the original manuscript we did not provide any clear explanation on how the 3D volume are estimated from the 2D model. This resulted in misleading the reviewer (and any other potential reader) in his estimate of the average thickness of our simulated dykes. In the revised version of the manuscript we explain how 3D volumes are estimated starting from 2D cross sections of dykes, and add the information of the thickness of our simulated dykes, lines **268-269 / 431-434** and **609-624** (please refer to the manuscript with highlighted changes).

We will report the same explanation here as answer to the reviewer concern:

The initial input for our model is the cross section of the intrusion, which is a “volume per unit length” along the out-of-plane direction. From this we estimate the 3D volume considering that the width of the intrusion (out-of-plane length) is of the same order of the in-plane dyke length (this information was missing). For instance, the smallest “volume per unit length” we tested, produce a dyke length (in-plane) of 3.2 km, for a cross section of $8 \cdot 10^{-4}$ km². Therefore, we estimate the volume as $(8 \cdot 10^{-4} \times 3.2)$ km³, assuming that the dyke width (out-of-plane) is equal to 3.2 km (in-

plane length). This results in the 3D volume given in the manuscript: $\sim 2.5 \cdot 10^{-3} \text{ km}^3$ (line 163). From this, one can estimate the average opening of our thinner intrusions being $\sim 0.25 \text{ m}$. Actually, the maximum opening of our thinner dykes (tear drop shaped) is $\sim 0.4 \text{ m}$. Such thickness would not pose drastic problems for propagation, at least for time scales of some days, which could be a reasonable assumption for the distances considered here. We would also like to stress that our estimates from the 2D cross sections do not represent the exact 3D critical volumes for propagation. In fact, our 3D volume estimation, based on the assumption of a constant width, may overestimates of the actual 3D volume, because of the expected rounded shape of the propagating tip of a 3D hydrofracture (cf. Watanabe et al., 2002). On the other end, by neglecting the magma loss in the dyke tale due to cooling and viscous effects, the critical cross section for propagation computed by our model may be underestimated. In addition, as discussed in the method section other parameters (which sometimes are difficult to constrain) would also affect the critical volume for propagation.

Minimum volume needed: In the same spirit as the previous point between 2D and 3D volume, I don't think this add much value to the paper. A more general discussion could be made on the fact that more volume/buoyancy is needed to reach the surface on one side of the volcano than for the other side. Stating that smaller volume is needed to reach the surface where indeed new eruptive activity is observed is probably enough. To strengthen the discussion, I would move lines 406 to 416 from "limitations in data and observables", into the main body of the paper where critical volumes are discussed (lines 297 to 303).

Reply: We agree with Benoit Taisne, therefore we moved part of the discussion for the critical volume for propagation to the "Results" section, and to the section "The dyke propagation model" in "Method", following Benoit Taisne's suggestion. Lines **270-275 / 434-438** (please refer to the manuscript with highlighted changes).

Few minor comments:

1) Regarding Fogo magmatic plumbing system, reference based on petrology would be useful. Constraining such a deep source using geodetic data only is not a strong argument (with respect to the size of the island itself...), at best you can probably rule-out shallow depth. I found a PhD dissertation, and abstract discussing the origin depth of the erupted magma, by Hildner Elliot, that is in accordance with the one given in the text (reference to be added line 106-107).

Reply: We thank Benoit Taisne for providing this reference to a petrological study. We added a sentence and the reference to Hildner et al. (2012) at lines **153-155** (manuscript with highlighted changes).

2) They have a paper on the 1951 pre-eruptive source and storage that would be relevant for this study. Last line of there abstract: "Thus petrologic data indicate that flank collapse events may ignificantly influence deep-seated magma plumbing systems beneath ocean islands." Hildner et al., 2012, Barometry of lavas from the 1951 eruption of Fogo, Cape Verde Islands: Implications for historic and prehistoric magma plumbing systems, JVGR. <http://dx.doi.org/10.1016/j.jvolgeores.2011.12.014>

Reply: We added the reference to Hildner et al. (2012) at line **155**.

3) I'm surprise by the comment line 417-422 on the possibility to include uncertainties to the results. How long does a run last? Once again, if the discussion on the volume remains qualitative and not quantitative this is not an issue anymore.

Reply: Benoit Taisne is actually correct, we modified our comment on the possibility to include

uncertainties in our simulations, lines **628-632** (manuscript with highlighted changes).

Typos/Edits:

Line 181 (reference to figure 4c): I would suggest adding a dashed line instead of mentioning the orange colour contour.

Reply: We added dashed lines in **Fig. 4c**, as suggested by Benoit Taisne.

Line 353: Boundary Elements instead of BE (line 68 is far at this point)

Reply: Fixed.

Figure 1) would gain by adding the 2D cross section considered for Fogo volcano, as well as adding the rift zone for Fogo.

Reply: We added the cross section of Fogo in **Fig. 1**.

Figure 3) x_c in the caption refer to x_e in the figure.

Reply: Fixed.

REVIEWERS' COMMENTS:

Reviewer #1 (Remarks to the Author):

Based on careful reading of this new version, I have seen that the authors amended the original manuscript by incorporating all suggestions and corrections. They now acknowledged previous fundamental literature that was not cited in the original version. Above all, they clearly stated which are their real new findings respect to similar previous works made by other authors. I think that now it can be published in Nature Communication, although I recommend to revise references, since there is some error: for example, the paper Acocella et al., 2005 cited in the text, does not exist in the reference list.

Reviewer #4 (Remarks to the Author):

Dear Authors,

Your manuscript presents an analysis of the possible outcomes on dyke propagation following a flank collapse. There are many parameters controlling the dynamics of the system and you make a good job at simplifying the problem and identifying the important controlling factors. You describe the previous contributions well and your study complements previous investigations on shallower processes. In particular, your study presents an important advancement on the methodology, as you present a technique to propagate dykes in an arbitrary stress field. Your model is limited to plane strain, but this simplification makes the study more digestible for a wider audience. Anyhow, the shortcomings of the model are well described and make an interested 'to do' list for the community. The paper is clearly suited for publication in Nature Communications after a few very minor revisions.

- A major concern is the applicability of the plane strain solution for surface loads. This is not described in the manuscript, but I suspect that the authors use the so-called Flamant solution which describes the effect of a singular load on the displacement and stress in the half space (half plane). This solution is a classic application of the Airy solutions described by Michell (1899). The authors may have simply computed the convolution between the singular solution with the shape functions described in Fig. 2. Quite notoriously, the Flamant solution predicts unbounded displacements away from from the surface load and this problem does not disappear with a finite source. This is quite problematic. A workaround is to consider the restoring forces from buoyancy, for example, due to the rigidity contrast at the surface or at the Moho. A solution in the Fourier domain is described in the work of

Luttrell, Karen, and David Sandwell. "Ocean loading effects on stress at near shore plate boundary fault systems." *Journal of Geophysical Research: Solid Earth* 115.B8 (2010).

Luttrell, Karen, et al. "Modulation of the earthquake cycle at the southern San Andreas fault by lake loading." *Journal of Geophysical Research: Solid Earth* 112.B8 (2007).

It is quite possible that this issue may not be significant in this case, as the wavelength is small, but this is worth checking.

- Line 333, is the viscosity of the magma simply (and only) influencing the time scale of the dyke propagation? Or does the viscosity actually affect the path of propagation?

- Line 399, can you identify the combination of parameters that control the path deviation, as opposed to a long list of independent parameters? I imagine that Kr/G , K/G , $Kr/(\rho g)$ are important non-dimensional parameters. What about viscosity?

- Line 130, "Constraints"

Point-by-point response to the reviewers

Manuscript: "The effect of giant lateral collapses on magma pathways and the location of volcanism"

Reviewer#1: *Based on careful reading of this new version, I have seen that the authors amended the original manuscript by incorporating all suggestions and corrections. They now acknowledged previous fundamental literature that was not cited in the original version. Above all, they clearly stated which are their real new findings respect to similar previous works made by other authors.*

I think that now it can be published in Nature Communication, although I recommend to revise references, since there is some error: for example, the paper Acocella et al., 2005 cited in the text, does not exist in the reference list.

Reply: We thank the Reviewer#1 for his/her further review. We are very happy that he/she finds that the manuscript has substantially improved and is now suitable for publication in Nat. Comm.. We also thank the reviewer for spotting some errors among the references, which we fixed in this revised version.

Reviewer#4: *Dear Authors, Your manuscript presents an analysis of the possible outcomes on dyke propagation following a flank collapse. There are many parameters controlling the dynamics of the system and you make a good job at simplifying the problem and identifying the important controlling factors. You describe the previous contributions well and your study complements previous investigations on shallower processes. In particular, your study presents an important advancement on the methodology, as you present a technique to propagate dykes in an arbitrary stress field. Your model is limited to plane strain, but this simplification makes the study more digestible for a wider audience. Anyhow, the shortcomings of the model are well described and make an interested 'to do' list for the community. The paper is clearly suited for publication in Nature Communications after a few very minor revisions.*

Reply: We thank the Reviewer#4 for his constructive comments, they particularly helped us to improve the discussion concerning the modelling approach that we used to simulate loading and unloading forces.

- A major concern is the applicability of the plane strain solution for surface loads. This is not described in the manuscript, but I suspect that the authors use the so-called Flamant solution which describes the effect of a singular load on the displacement and stress in the half space (half plane). This solution is a classic application of the Airy solutions described by Michell (1899). The authors may have simply computed the convolution between the singular solution with the shape functions described in Fig. 2. Quite notoriously, the Flamant solution predicts unbounded displacements away from the surface load and this problem does not disappear with a finite source. This is quite problematic. A workaround is to consider the restoring forces from buoyancy, for example, due to the rigidity contrast at the surface or at the Moho. A solution in the Fourier domain is described in the work of:

Luttrell, Karen, and David Sandwell. "Ocean loading effects on stress at near shore plate boundary fault systems." Journal of Geophysical Research: Solid Earth 115.B8 (2010).

Luttrell, Karen, et al. "Modulation of the earthquake cycle at the southern San Andreas fault by lake loading." Journal of Geophysical Research: Solid Earth 112.B8 (2007).

It is quite possible that this issue may not be significant in this case, as the wavelength is small, but this is worth checking.

Reply: We agree with the reviewer that the plane strain solutions we have been using in the present study (and that has been widely used for similar applications) present some peculiarities that need to be addressed. Following the reviewer comment, we further developed the discussion in section “Methods” - “Limits of the model and applicability”, lines 462-469 and 515-522 of the revised manuscript (please refer to the version with highlighted changes). The same discussion is reported here as answer to the reviewer comment.

In order to compute the effect of loading and unloading force distributions acting at the ocean floor, we considered the formulas given in *Davis and Selvadurai (1996)*, for normal tractions applied over a stripe of a certain length (plane strain approximation based on Flamant solutions, as pointed out by the reviewer). We superpose these stripe load solutions to describe different loading and unloading functions. As highlighted by the reviewer, the displacement field for these solutions has the property of becoming unbounded at infinite distances from the line of applied load (cf. eq. 13.30 pag. 407, in *Jaeger et al., 2007*). *Davis and Selvadurai (1996)* showed that this unrealistic result can be overcome by working with the relative vertical displacements between two points on the surface (*Jaeger et al., 2007, pag. 406*). However, in our application, we do not consider the displacement induced by loading/unloading forces, since we are only interested in the stress perturbation induced within the crust. We discussed the implications of the plane strain approximation at line 462-469.

In addition, as pointed out by the reviewer, a possible approach to overcome the problem of unbounded displacements is the one used by *Luttrell et al. (2007; 2010)*. *Luttrell et al. (2007)* computed the Coulomb stress change on the southern San Andreas fault, induced by changes in the water level at lake Cahuilla. They considered changes in the pore pressure, the effect of elastic bending of the lithosphere, and its viscoelastic relaxation. They found that the joined effect of pore-pressure and viscoelastic bending causes relevant changes of normal stress on the fault plane at seismogenic depth. A similar approach is used by *Luttrell et al., 2010* to compute the effect of eustatic sea level changes on near shore plate boundaries. They found that the plate bending in response to sea level changes alter the state of stress within the lithosphere within a half flexural wavelength of the coast.

The effect of lithosphere bending beneath and close to the edges of the loaded region causes further horizontal compression at the top of the plate and extension at his bottom (see for instance Fig. 3 in *Luttrell et al., 2007*). The opposite (shallow extension and deep compression) is predicted on the sides of the loaded region. As the reviewer pointed out, the effect of lithosphere bending depends on the wavelength of the loading, the lithosphere thickness, and the distance at which the effect of the loading is considered. In our application, we focus in an area centred below the loading region, relatively far from the edges of the volcanic loading. In such conditions we expect bending stresses to be less important than the ones obtained by *Luttrell et al. (2007; 2010)*. Nevertheless, bending of the lithosphere below the loading region may still add horizontal extension at Moho depth and further compression in the shallow crust (*Watts, 2001*). Therefore, we would expect that the bending of the lithosphere may deflect dyke trajectories further towards the sides of the volcano, magnifying the effect of collapse unloading: In fact, horizontal extension at Moho depth, would favour the initial vertical rise of dyke intrusions, while increased horizontal compression in the shallow crust would enhance the outwards bending of the maximum compressive stress direction (which in our stress model happens only as a result of the unloading induced by the collapse).

Finally, a more complex stress model for the response of the lithosphere to loading and unloading forces due to volcano growth and destruction would certainly represent a step forward with respect to our simplified plane strain stress model. For this reason, we actually consider the coupling of our model for dyke propagation with more complex and realistic stress models as one of the priorities of our future research. Nevertheless, we would like to highlight that the stress model we implemented for the present study, is able to capture the importance of the trade-off between volcanic loading and collapse unloading.

- Line 333, is the viscosity of the magma simply (and only) influencing the time scale of the dyke propagation? Or does the viscosity actually affect the path of propagation?

Reply: This is a very interesting point. The question raised by the reviewer is actually still unanswered, at the current state of research. None of the currently existing models simulating dyke paths consider the viscosity of magma, while those models that take into account magma viscosity are not able to compute dyke trajectories. *Pinel et al. (2017)* is the first attempt of joining different modelling approaches of dyke propagation by accounting for both, magma viscosity and complex trajectories of intrusions. However, while the study by *Pinel et al. (2017)* addresses the velocity of dyke intrusions along different complex trajectories, it does not include the effect of viscosity on the trajectory calculation. We expect that intrusions with higher viscosity will be less sensible to external stress changes, and show less deflection of their trajectories, since viscous magma requires higher overpressure to propagate: In fact, the tendency of a dyke to follow the trajectory defined by the external stress field decreases with increasing dyke overpressure (cf. *Watanabe et al., 2002*).

- Line 399, can you identify the combination of parameters that control the path deviation, as opposed to a long list of independent parameters? I imagine that Kr/G , K/G , $Kr/(\rho g)$ are important non-dimensional parameters. What about viscosity?

Reply: This is another very interesting point. The tendency of a dyke (or hydrofracture) to orient perpendicular to the least compressive stress depends on the interplay between the magnitude of the deviatoric component of the external stress and the overpressure (or driving pressure) of the intrusion (*Watanabe et al., 2002*, *Maccaferri et al., 2011*). *Watanabe et al. (2002)*, for instance, find a threshold of ~ 0.2 for the ratio between the shear stress on the crack plane and the fluid overpressure in order for a dyke to deviate its path, i.e. below this value no deflection of the intrusion is expected. Our model generally confirms this experimental observation (*Maccaferri et al., 2011* and *Pinel et al., 2017*). However, other parameters may affect the likelihood of an intrusion to be deflected by an external stress field, such as the length of the intrusion, which would affect the stress intensity factor that is associated with the fluid-filled crack under the condition of constant fluid overpressure. Therefore, it is not trivial to identify a unique non-dimensional parameter that controls the dyke deflection. We are addressing this interesting problem in ongoing research by comparing our numerical model with analogue experiments. Hopefully we will soon be able to present first results. As for the effect of magma viscosity, we refer to the answer to the previous point of the reviewer.

- Line 130, "Constraints"

Reply: Comment accepted and changes made.

References:

- Davis, R. O., and A. P. S. Selvadurai. 1996. *Elasticity and Geomechanics*. Cambridge University Press. Cambridge.
- Jaeger, J. C., Cook, N. G. W., Zimmerman, R. W., 2007 *Fundamental of rock mechanics*, Fourth edition, Blackwell publishing.
- Luttrell, Karen, and David Sandwell. Ocean loading effects on stress at near shore plate boundary fault systems. *Journal of Geophysical Research: Solid Earth* 115.B8 (2010).
- Luttrell, Karen, et al. "Modulation of the earthquake cycle at the southern San Andreas fault by lake loading." *Journal of Geophysical Research: Solid Earth* 112.B8 (2007).

- Maccaferri, F., Bonafede, M. and Rivalta, E., 2011. A quantitative study of the mechanisms governing dike propagation, dike arrest and sill formation. *Journal of Volcanology and Geothermal Research*, 208(1–2), pp.39–50.
- Pinel V, Carrara A, Maccaferri F, Rivalta E, Corbi F. (2017) A two- step model for dynamical dike propagation in two dimensions: Application to the July 2001 Etna eruption. *Journal of Geophysical Research: Solid Earth*. 1;122(2):1107-25.
- Watanabe, T., Masuyama, T., Nagaoka, K. and Tahara, T., 2002. Analog experiments on magma-filled cracks. *Earth, planets and space*, 54(12), pp.1247-1261.
- Watts, A. B., 2001. *Isostasy and Flexure of the Lithosphere*. Cambridge University Press, Cambridge, ISBN 0521006007.